# Eye movements during free viewing to maximize scene understanding

Shravan Murlidaran [1,4] ✉ & Miguel P. Eckstein [1,2,3,4]

What humans look at and do when freely viewing a scene is not well understood. We measure observer eye movements under different instructions while participants view a customized set of image pairs containing small visual alterations that greatly change scene interpretation (Winograd images). We show that free-viewing fixations resemble those of observers describing scenes but differ from those of observers counting or searching for objects. Fixations are more often directed toward people and objects whose removal most alters scene interpretation, rather than toward the most salient or meaningfully judged regions (meaning maps), or objects perceived to be grasped or gazed at. Small image changes that modify scene understanding (Winograd images), but not salience or meaning maps, alter fixation patterns. By instructing observers to describe scenes while fixating on objects either relevant or irrelevant to scene understanding, we demonstrate that free-viewing eye movements are functionally important for accurate scene comprehension. Thus, an important human default task of free viewing eye movements is to comprehend scenes.

Eye movements are a critical part of human vision. Ever since Buswell (1935)[1] and Yarbus (1967)[2] conducted their classic eye-tracking studies, researchers have tried to understand what drives and influences eye movements. One of the prominent theories proposed two decades ago is that people make eye movements to salient regions. Saliency was originally defined (Itti et al.[3]) in terms of a combination of low-level features such as color, intensity, and orientation at any given location relative to its immediate surroundings, with many subsequent elaborations of the model[3-8]. Since then, studies have shown that when humans perform specific tasks such as searching for an object or person[9-13], executing a motor action[14-16], navigating in an environment[17,18], or identifying faces[19], they do not look at the most salient object/region. Instead, humans look at locations that contain objects or visual features relevant to the task[20-31], and/or that allow maximizing task accuracy[19,32,33]. When instructed to artistically evaluate or describe scenes using keywords, observers fixate on objects more than salient regions[34].

Humans are not always engaged in specific tasks. They might look through a bus window, sit on a park bench, wait for a restaurant table,

and explore a scene with no particular task. This is often called free viewing[4,35]. Fixations during free viewing have been commonly used to support the theory that people direct eye movements to salient regions[36-42]. More recently, studies have shown that even during free viewing, people do not direct their eyes to salient regions but rather to objects[43] or meaningful regions measured by observers' subjective judgments of the meaningfulness of segmented local patches of scenes (meaning maps; subsequently referred to as locally meaningful regions)[44,45].

Here, we introduce a new theory. We hypothesize that, during free viewing, humans aim to understand a scene. To rapidly do so, they direct their eyes to regions critical to scene understanding rather than looking at those salient or segmented regions, locally judged to be meaningful. An important distinction between a locally meaningful region and one critical to scene understanding is that the latter considers how the region, people, or object contributes to understanding the whole scene. For example, Fig. 1a illustrates the concept with an image described by observers as: Someone replacing batteries of a TV

¹Psychological & Brain Sciences, University of California, Santa Barbara, Santa Barbara, CA, USA. ²Department of Electrical and Computer Engineering, University of California, Santa Barbara, Santa Barbara, CA, USA. ³Department of Computer Science, University of California, Santa Barbara, Santa Barbara, CA, USA. ⁴These authors contributed equally: Shravan Murlidaran, Miguel P. Eckstein. ✉e-mail: smurlidaran@ucsb.edu

remote control. The highly contrasting blue paper is the most salient object, and the sunglasses are the most locally meaningful object. However, the inverted remote with its battery cap open is the most critical object to the scene's understanding. How would one measure the contribution of an object to the scene's understanding? We propose a new method by having observers describe the scene and assess the impact of removing one object at a time on the scene descriptions. Removing the remote control from the image in Fig. 1a significantly alters the description of the scene, whereas eliminating the sunglasses or the blue paper does not (see Fig. 1a, right side). Thus, our theory hypothesizes that the remote control, critical to scene understanding, would attract more human fixations than the blue paper or the

sunglasses. Furthermore, our second hypothesis is that fixating the object critical to scene understanding has a functional role in extracting the information required to understand it. The theory would predict that directing the high-resolution fovea to the remote control would be important to understand the scene (accurately describe it), while fixating on the blue paper would not. Although previous work has related eye movements to scene comprehension when observers are instructed to describe scenes[46–48], these studies have not examined how free-viewing fixations relate to the contributions of each object to scene understanding, evaluated other tasks or models, or experimentally investigated the causal functional importance of fixations on scene understanding.

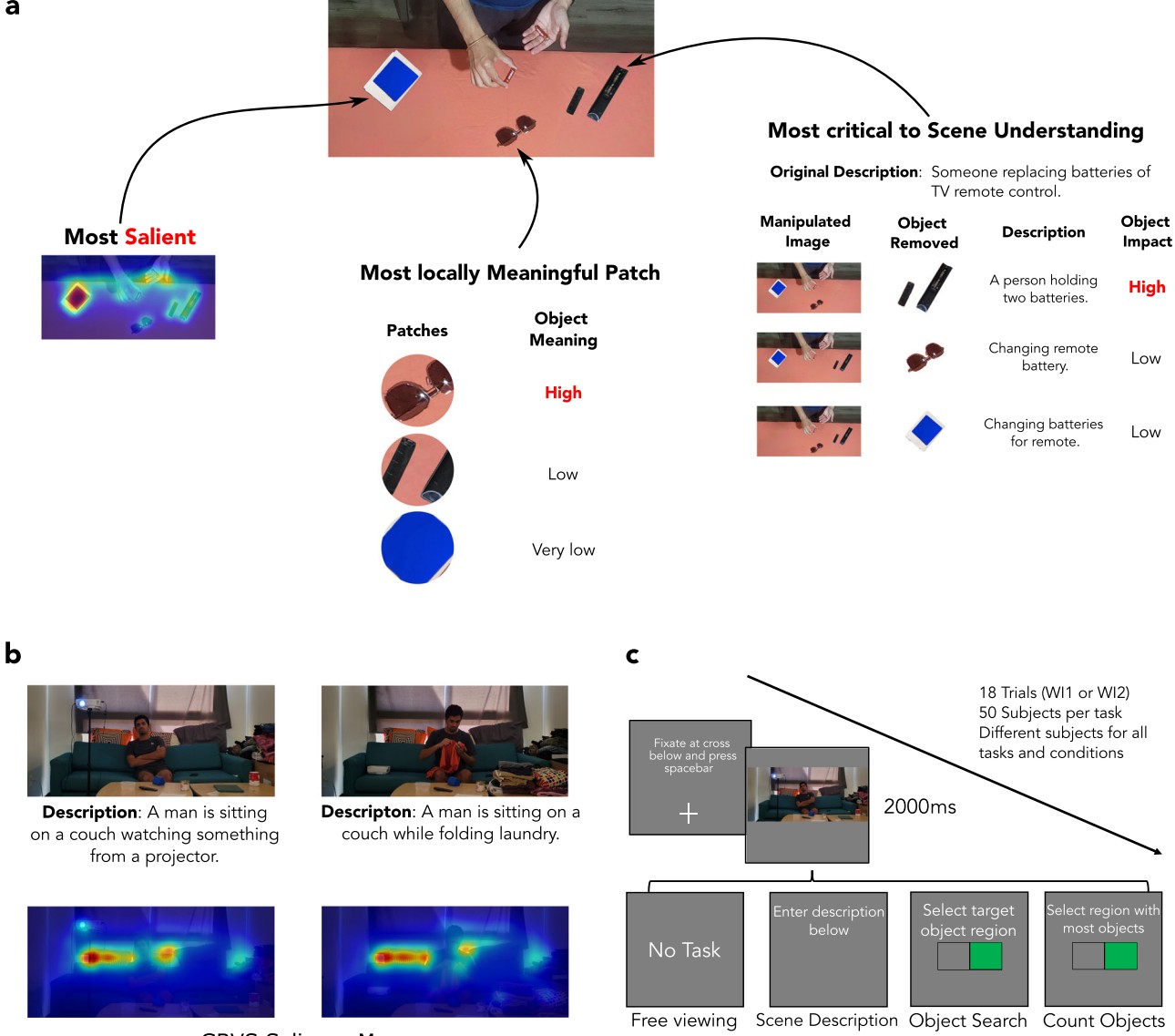

**Fig. 1 | Motivation and experimental design. a** Example of a scene where the object corresponding to the most locally meaningful patch/region (meaning maps), the most salient object, and the object most critical to understand the scene are different. The most salient object is the contrasting blue paper, while the most locally meaningful patch corresponds to the sunglasses. The image is described as: Someone replacing batteries of TV remote control. The hands, the batteries, and the TV remote control are objects critical to understanding the scene, while the salient blue paper and sunglasses are irrelevant to the scene's meaning. This image is not part of the study but included as a simple example to illustrate the concept of objects critical to scene understanding. **b** First row: Winograd image pair created to

dissociate saliency from relevance to scene understanding with corresponding descriptions from observers. The projector is relevant to the scene understanding for the left image, while the clothes are relevant for the right image. Second row: The prediction of the GBVS saliency model for both Winograd images is similar, even though the descriptions and objects relevant to scene understanding vary. **c** Procedure for the four experimental conditions (free viewing, scene description, object search, and counting objects) of the eye movement study with 50 participants per condition (between-subjects design). Consent was obtained from the people in the featured image for its publication.

Our initial methodology and analysis focused on predicting human fixations on different objects. In the latter part of the paper, we generalize the analysis to assess whether the theoretical framework can predict the well-documented frequent fixations on people (bodies and faces[49,50]; for example, deleting the hand with the batteries in Fig. 1a would maximally alter the description of the scene). We also assess how social cues (gaze and grasp cues) fit the theoretical framework.

One of the main challenges in evaluating the hypothesis that eye movements are directed to people or objects that maximize scene understanding is that in most image data sets[51–56] the person or object critical to scene understanding is often also the most salient object and the locally most meaningful region. This is a consequence of photographers' practice of typically positioning the people or objects critical to the scene's understanding as salient and central in the photograph[57]. To overcome this challenge, we developed a new dataset that decouples salient regions from regions that contribute to scene understanding.

We created pairs of real scenes that were minimally altered visually but resulted in large changes in the scene's understanding (Fig. 1b; left) while maintaining the saliency maps of the scenes and the local meaningfulness of each region. These small visual changes across image pairs could be manipulations of an object's position, its substitution by another object, or an actor's posture. We refer to these image pairs as Winograd Images (WI), inspired by the Winograd schema developed for sentences for which minor modifications of a word in a sentence lead to a large change in the sentence's meaning[58]. To determine which objects were critical to the scene's understanding, we assessed how deleting each object impacted the accuracy of scene descriptions relative to the intact images.

To test our first hypothesis, we had different groups of observers view our Winograd pair image dataset following different condition instructions (experiment 1): search for a specific object (object search; OS), describe the scene (scene description; SD), count the number of objects (counting objects; CO), or critically, free viewing (no instructions; FV). If observers execute eye movements during free viewing to understand the scene, we would expect various specific results. First, fixation patterns during free viewing should be similar to those obtained from observers during the scene description condition, which explicitly requires the observer to understand a scene to describe it. Fixation locations during free viewing should be less similar to the fixations of observers executing other specific tasks that do not require understanding the scene, such as searching for an object or counting objects. Second, Winograd image pairs, which alter scene understanding, should result in varying fixation patterns (across Winograd images) for the scene description and free viewing conditions, but less for the object search and counting conditions. Third, fixations during the free viewing and scene description conditions should be directed to objects critical to scene understanding. In contrast, fixations during the object search would be directed to the object searched (target), and fixations during object counting should be equally spread among objects. As benchmark comparisons, we also evaluated the ability of a saliency model (Graph-Based Visual Saliency (GBVS)[4]), a human eye movement-trained deep neural network (DeepGaze[59]), and measurements of the most meaningful local regions (meaning maps[44]) to predict the fixations across Winograd pairs. We hypothesize that the objects critical to scene understanding in our data would be fixated more frequently than the most salient or locally meaningful region (as measured by meaning maps) for the free viewing and scene description conditions.

To test the functional role of fixations during free viewing in scene understanding, we conducted a second experiment. Observers described a scene while maintaining fixation at an object critical to scene understanding or another object irrelevant to scene comprehension. Suppose that fixations play a functional role in rapidly extracting the information necessary to understand a scene. In that case, scene descriptions should be more accurate when observers maintain fixation on objects relevant to scene understanding than when they fixate on objects irrelevant to comprehending the scene.

We show that observers' free-viewing fixations are more similar to fixations of observers instructed to describe the scenes than observers' fixations when counting objects or searching for specific objects. Small visual alterations to images that modify a scene's understanding but not the most salient or its meaning map (Winograd image pairs) change where humans most frequently fixate. Free viewing fixations are more frequently directed to people and objects critical to the understanding of a scene (elements that, when erased from the scene, maximally alter the scene's description) rather than the most salient, most meaningfully judged scene region (meaning map), or the object perceived to be grasped or gazed at. The theoretical framework also explains the higher frequency of fixations on people than on objects for most scenes because, when people are erased, scene descriptions are maximally altered. A temporal analysis shows that the first few fixations are most frequently directed to people and objects perceived to be grasped or gazed at, while later fixations increasingly focus on people and objects relevant to scene understanding. We also show that observers' scene description accuracy is higher while maintaining fixation on objects relevant than on objects irrelevant to scene understanding, suggesting that eye movements during free viewing are functionally important to comprehend scenes accurately.

## Results

### Free viewing fixations are most similar to scene description

We created a total of 18 Winograd pairs[60]. Fifty participants took part in each of the four conditions (a between-subjects design, with a total of 200 participants). Within each condition, participants were assigned randomly and equally to view one set of Winograd pairs (18 image trials). Figure 1c details the procedural flow of the four conditions.

Our primary analyses removed fixations/model predictions on human faces or bodies in the scenes. This pre-processing enabled us to focus on fixations on objects that were critical versus those irrelevant to scene understanding, rather than on human figures, which were critical across all Winograd image pairs. Figure 2a shows an example of the fixation heat maps (see methods for details) generated using 25 participants for each of the four conditions. The gray regions indicate fixations on people that were not considered in our primary analysis (but see further below for an analysis including fixations on people). Figure 2b shows the correlation of fixation heat maps across conditions (for 25 observers) computed for each of the 36 images (18 pairs) in our dataset. The fixation heat maps for free viewing were significantly more similar to the fixation heat maps from the scene description condition (FV-SD, r = 0.54) than to those from the object search (FV-OS, r = 0.32, bootstrap resampling of observers and images for all analyses, $p < 0.001$ vs. FV-SD, Cohen's d = 1.28) or counting objects (FV-CO, r = 0.42, $p < 0.001$ vs. FV-SD, Cohen's d = 0.75, see Figure Supplementary 1a). To compare the across-condition fixation heat map correlations to an upper bound, we estimated a within-condition (FV-FV) correlation of fixation heat maps, each using distinct subgroups of n=12 observers (Fig. 2c, see methods). As with the n=25 analysis, we found higher FV-SD fixation heat map correlations (r = 0.4) than those for FV-OS (r = 0.23, $p < 0.001$, Cohen's d = 1.19) and FV-CO (r = 0.29; $p < 0.001$, Cohen's d = 0.9). Notably, the FV-SD fixation heat map correlation was the closest to the within-condition (FV-FV) inter-observer (see dotted lines in Fig. 2c) heat map correlations ($\Delta r = r_{FV-FV} - r_{FV-SD} = 0.06$) relative to the other condition comparisons (the FV-OS vs. FV-FV correlations: $\Delta r = 0.26$; and the FV-CO vs. FV-FV correlations: $\Delta r = 0.17$).

### Changes in scene understanding with small visual manipulations influence observer fixations

We compared the measured human fixation patterns across the Winograd image pairs for different conditions with other existing

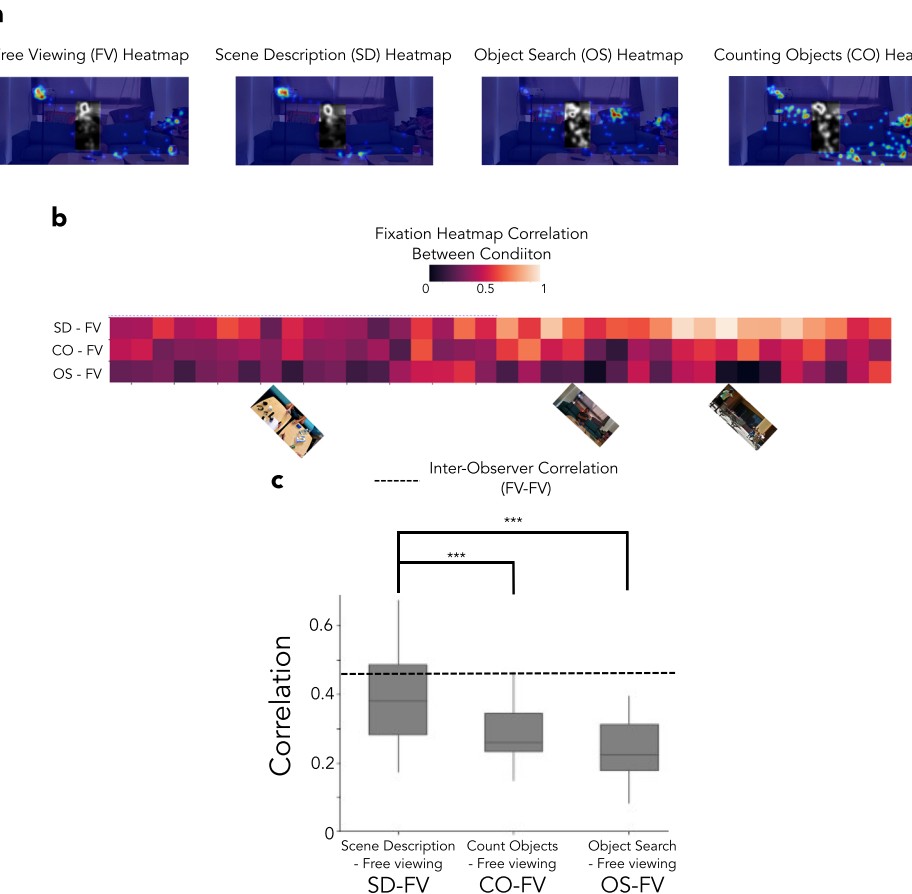

**Fig. 2 | Fixation heat map correlations across conditions. a** Example of the fixation heat maps generated from 25 participants for all four conditions. The gray heat map regions indicate regions containing people and were excluded from the initial fixation analyses. **b** Correlation between free-viewing fixation heat maps and fixation heat maps from other conditions for each image in our dataset (36 images). The lighter the color, the higher the correlation. Three sample images are shown. There was a trend of higher correlations between the free viewing (FV) fixation heat maps and those of the scene description (SD) condition than with the heat maps of the object search (OS) and counting objects (CO) conditions. **c** The average correlation between the free-viewing fixation heat maps and scene description fixation heat maps (12 participants and 36 images) was significantly higher than the correlation with the fixation heat maps of the OS and CO conditions ($p < 0.001$). The dotted line shows the inter-observer correlation of fixation heat maps from groups of 12 observers; FV-FV. (***$p < 0.001$). A one-tailed bootstrapped analysis was conducted to test the significance of all the results. In **c**, the line within the box indicates the median. The box spans the interquartile range (IQR), and the whiskers extend to the most extreme values within 1.5 × IQR across images. Consent was obtained from the person in the featured image for its publication.

fixation prediction models: a purely low-level saliency model (GBVS)[4], a neural network model trained on human fixations and images to predict fixations on novel images (DeepGaze II)[59], meaning maps that quantify the meaningfulness of local regions of scenes[44], and our newly proposed scene understanding maps that quantify the contribution of each object to the meaning of the entire scene. Figure 3a, b provides a flow chart of all the models we have used in this comparison.

To measure the contribution of an object to scene understanding, we developed a technique that measures the impact of an object present in the scene on the description provided by participants. Figure 3a illustrates the procedure with an example. The original descriptions for the intact (i.e., no object removed) images were compared with the descriptions of the images after the object removal to quantify the object's impact on scene understanding. If the object is irrelevant to scene understanding, the scene descriptions with the object present or removed would be similar. In contrast, if the object is critical to scene understanding, removing it would greatly impact the scene descriptions. The similarity of the scene descriptions can be measured by relying on a separate group of participants who view sentence pairs (without the image) and rate their similarity. Quantifying the impact of removing each object on the scene description allows us to create a map that visualizes the objects' contribution to scene understanding (scene understanding map, SUM). The SUM can be used to predict fixation distributions and the most fixated object (see methods and Fig. 4a). Similar SUM results were obtained using similarity ratings of descriptions based on automated Large Language Models (LLMs) based metrics (cosine similarity computed with the LLM embeddings of the sentences) rather than human ratings. The human-LLM rating agreement ($r_{Gemini-Human} = 0.7$, $r_{GPT4-Human} = 0.72$) was comparable to the average human-human ($r = 0.75$) agreement in the similarity ratings of these descriptions (see Figure Supplementary 1b, c).

We hypothesized that altering the scene understanding through small visual manipulations would change the human fixations in both the free viewing and scene description conditions. Thus, we predicted that the correlation of human fixation distributions across Winograd image pairs would be smaller than that for the saliency, local meaningful regions, and DeepGaze prediction maps. Similar to the human fixation distributions, our hypothesis predicts that the correlation across the scene understanding maps of Winograd image pairs would also be low. In addition, human fixation heat maps would be more similar (higher correlation) across the Winograd image pairs for the object search and counting objects conditions compared to the free viewing and scene description conditions.

Figure 4a shows examples of the predicted fixation heat maps of different models and measured fixation heat maps for humans for the

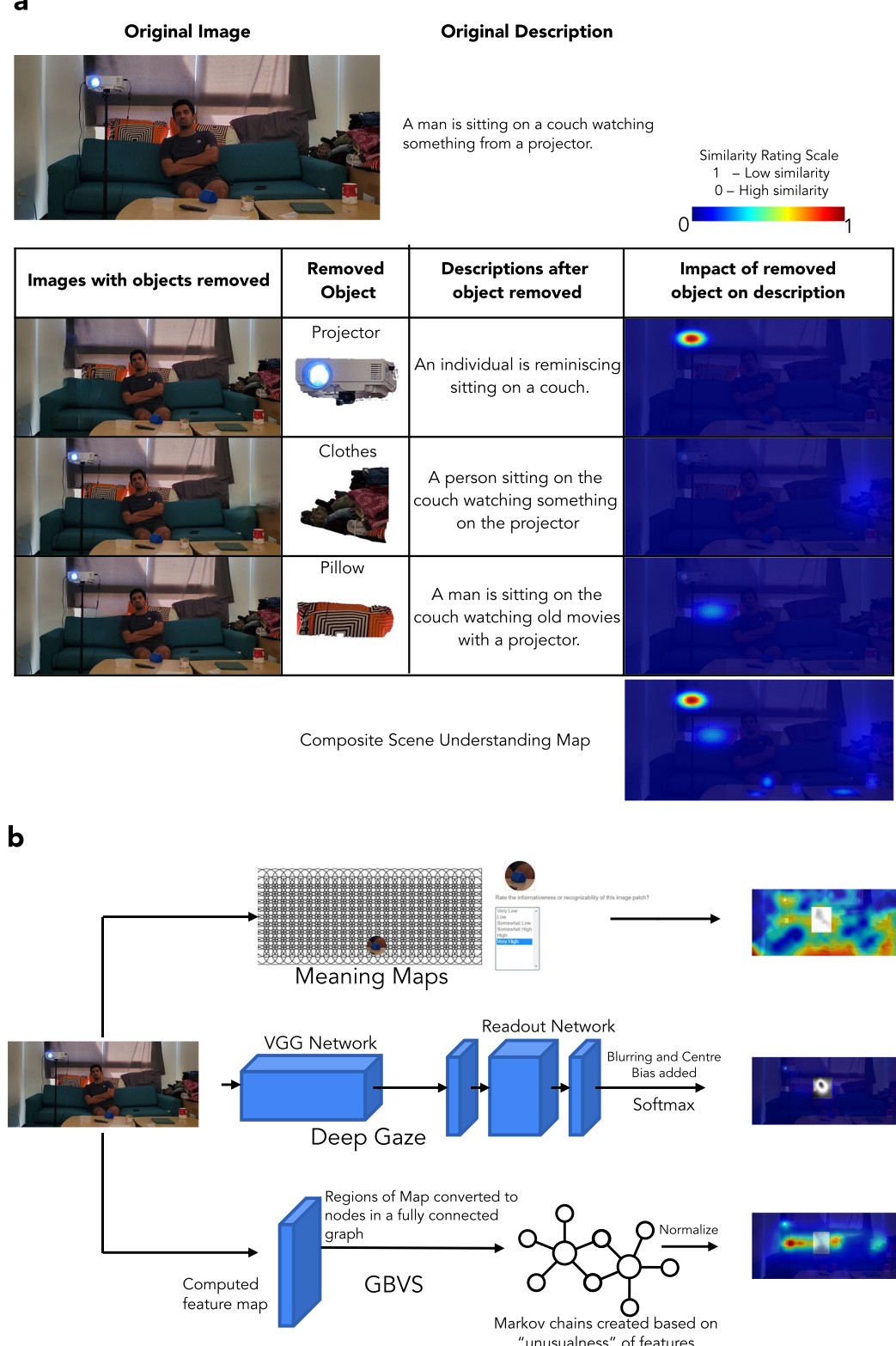

**Fig. 3 | Scene understanding maps and other fixation prediction models.**
**a** Procedural workflow for creating our proposed scene understanding maps (SUM): The impact of an object on scene understanding was quantified by comparing the similarity (rated by humans) between descriptions that people provide when the object was present and those when the object was removed. See Figure Supplementary 1b, c for details on the consistency of LLMs with human raters on rating the similarity of the descriptions. The SUM was generated by placing a 2D Gaussian on each object, with its amplitude determined by the impact of the object on the scene understanding and with a standard deviation determined by the size of the bounding box around the center of each object (SD range: 0.5 to 1.5 dva) ([43], see methods for details). **b** Workflow for creating maps for other fixation prediction models used in this study: meaning maps (top), DeepGaze (middle) maps, and GBVS (bottom) maps The grey regions in the heat maps indicate the predictions that fall on people in the scene and were excluded from the initial analyses (see further below for results that include fixations on people). Consent was obtained from the person in the featured image for its publication.

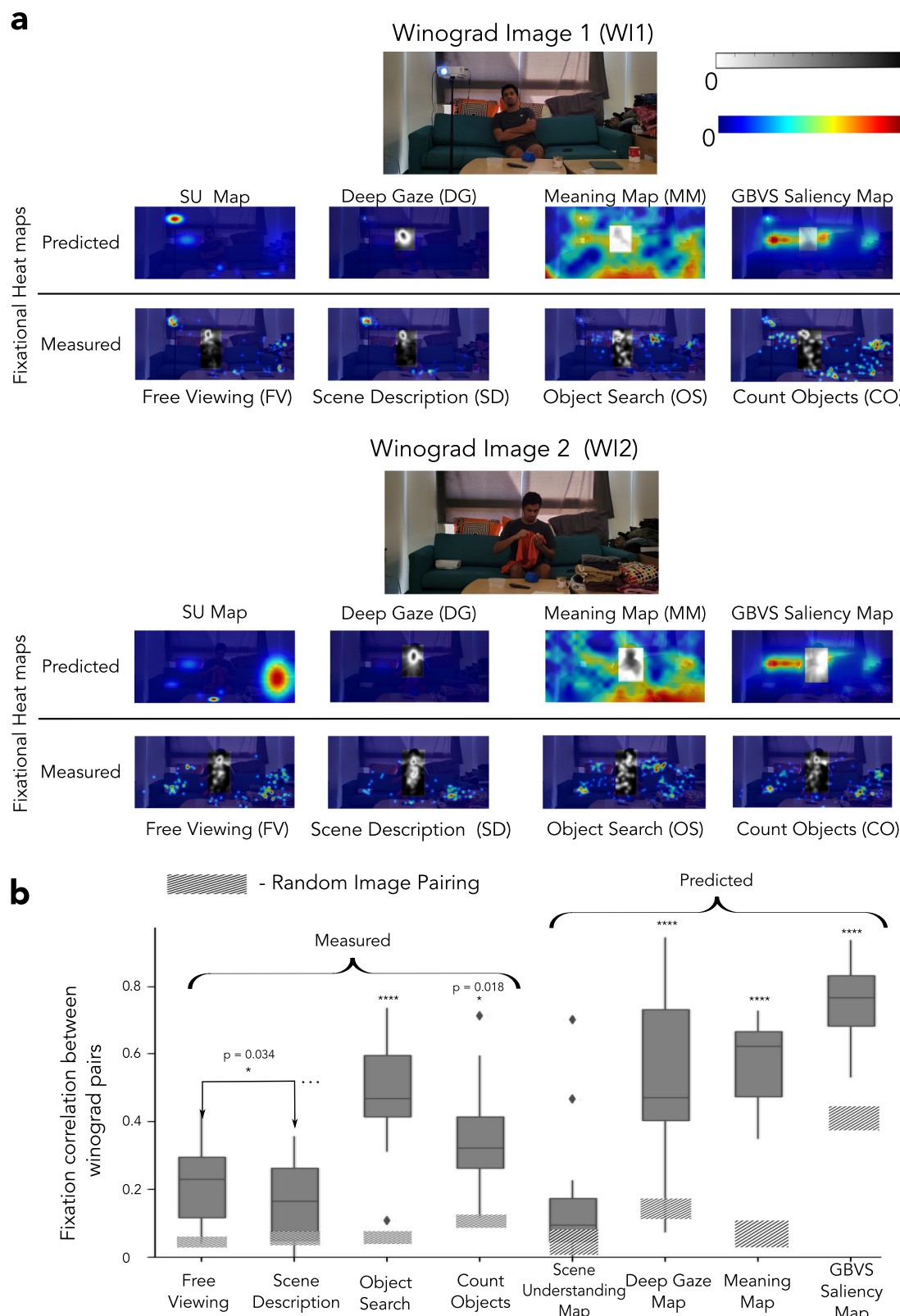

four conditions for a Winograd image pair. The correlation between the human fixation heat maps of Winograd images for the free viewing condition (Fig. 4b) was significantly lower than for the object search ($p < 0.001$, Cohen's d = 1.43) and counting objects ($p = 0.018$, Cohen's d = 0.93) conditions. The correlation of Winograd image fixation heat maps for the scene description condition was significantly lower than for the free viewing condition ($p = 0.034$, Cohen's d = 0.53). The

correlations of fixation prediction heat maps between Winograd images for saliency, meaning maps, and DeepGaze were significantly higher ($p < 0.001$ vs. free-viewing for all comparisons, Cohen's d = 1.86, 1.68, 1.25, respectively) than the correlations observed for free-viewing human fixation heat maps between Winograd pairs. In contrast, the correlation of the fixation predictions of Winograd image pairs for the SUM was the lowest and not different from that observed for human

**Fig. 4 | Measured/Predicted fixation heat map correlations across Winograd pairs. a** Examples of measured and predicted fixation heat maps for a Winograd image pair. The grey regions in the heat maps indicate the fixations/ model predictions that fall on people in the scene and were excluded from the primary analyses (see further below for results with fixations/model predictions to people included). **b** The correlation between Winograd pairs for free-viewing fixation heat maps (25 participants per image across 18 pairs of Winograd images) was low, suggesting that people look at different locations for each Winograd pair. Similar low correlations result from the scene description condition. The correlations between fixation heat maps for Winograd image pairs in the object search ($p <$ 0.001) and counting objects (p = 0.018) conditions were significantly greater than those for free-viewing. None of the existing fixation prediction models (GBVS saliency, meaning maps, and DeepGaze) predicts the low correlation observed in free viewing ($p < 0.001$). However, the scene understanding maps showed a low correlation across the Winograd image pair. The shaded area for each bar shows the correlations observed for random image pairing (rather than Winograd pairs). The box plot shows an image-level distribution of the Winograd correlations, with the middle line indicating the median, the box width indicating the IQR, and the whiskers indicating the 1.5*IQR. (****$p < 0.0001$; * $p < 0.05$). A one-tailed boot-strapped analysis was conducted to test the significance of all the results. In **b**, the line within the box indicates the median. The box spans the interquartile range (IQR), and the whiskers extend to the most extreme values within $1.5 \times$ IQR across images. Consent was obtained from the person in the featured image for its publication.

fixations for the free viewing and scene description conditions (p = 0.19, Cohen's d = 0.54 for free viewing; *p* = 0.51, Cohen's d = 0.12 for scene description condition).

These variations in Winograd pair fixation heat map correlations across conditions cannot be accounted for by differences in inter-observer variability ($r_{\text{inter-observer}} = 0.45$ for free viewing; $r_{\text{inter-observer}} = 0.47$ for scene descriptions; $r_{\text{inter-observer}} = 0.49$ for object search; $r_{\text{inter-observer}} = 0.38$ for counting objects condition, see Figure Supplementary 2a, dotted lines). In addition, as a control comparison, the correlations observed for a random pairing of images (shaded area in Fig. 4b) do not result in the ordering observed in Winograd image correlations across conditions.

Even though the SUM is most helpful in predicting the most frequently fixated object, it also attains comparable accuracy at predicting all human fixations to the meaning maps and DeepGaze and is better than GBVS (shuffled AUROC analysis; see methods for implementation details and see Figure Supplementary 2b for the results).

## Observers look at regions/objects critical to scene understanding during free viewing

To assess where human observers fixate most frequently during free viewing, we categorized objects based on their contribution to the scene understanding map (see Fig. 3a). We categorized objects as the most relevant to scene understanding (SU-relevant) if erasing them from the scene resulted in the largest change in the participants' scene description. Similarly, we categorized objects as irrelevant to scene understanding (SU-irrelevant) if erasing them from the scene did not result in large changes in the participants' scene description (see methods; this analysis also excluded people in scenes, see further below for analysis including people). To control for low-level visual properties of objects and object types, we designed our stimuli so that the same objects were SU-relevant for one Winograd image (example: projector in Fig. 5a; left, clothes in Fig. 5a; right) and SU-irrelevant to the complementary Winograd image (example: clothes in Fig. 5a; left, projector in Fig. 5a; right). Across the Winograd images, SU-relevant and SU-irrelevant categories were the same set of objects, providing a strong control.

We measured the frequency of human fixations on SU-relevant and SU-irrelevant objects for each condition. We also quantified the frequency of human fixations on the top predictions of the GBVS saliency model, DeepGaze, and the most locally meaningful region based on meaning maps. Figure 5b shows the fixation frequency for each object category averaged across all images and participants. Results are shown for the four conditions. Observers fixated on objects more frequently when they were critical to scene understanding than when they were irrelevant for both the free viewing (SU-relevant vs. SU-irrelevant objects, $p < 0.001$, Cohen's d = 1.41, adjusted p-values (q-value) reported for False Discovery Rate (FDR) at $\alpha = 0.05$ for 28 comparisons; 20 comparisons in Fig. 5b and four comparisons in Fig. 6d and four comparisons in Fig. 8c) and scene description conditions ($p < 0.001$, Cohen's d = 1.70). Furthermore, in the free viewing and scene description conditions, the SU-relevant objects were fixated

significantly more often than any of the top GBVS salient ($p < 0.001$ for both conditions, Cohen's d = 1.41, 1.46, respectively), DeepGaze (p = 0.005, Cohen's d = 0.90 and $p < 0.001$, Cohen's d = 0.98 respectively), or locally meaningful objects ($p < 0.001$ for both conditions, Cohen's d = 1.46, 1.51, respectively). In the search condition, the searched target was fixated the most frequently ($p < 0.001$, Cohen's d > 1.50 for all comparisons). Still, a smaller but significant difference was observed in fixation frequency between SU object categories (higher frequency to SU-relevant vs. SU-irrelevant objects) for the object search ($p = 0.048$, Cohen's d = 0.150) and the counting conditions ($p = 0.029$, Cohen's d = 0.790).

To assess how observer fixation preferences develop through time, we analyzed the cumulative fixation frequencies to SU-relevant and SU-irrelevant objects for all conditions as a function of fixation number after image onset. For the free viewing and scene description conditions, the cumulative fixation frequency difference across SU-relevant and SU-irrelevant objects ($\Delta CF_{R,I}$) became significant after the 3rd fixation ($p < 0.001$, Cohen's d = 1.24, 1.29, respectively, Figure Supplementary 3a, adjusted p-values (q-value) reported for FDR at $\alpha = 0.05$ for all 40 comparisons in Figure Supplementary 3a). No statistically significant difference was observed for the search and counting objects conditions up to the 10th fixation. Figure 5c compares $\Delta CF_{R,I}$ as a function of fixation number for all conditions. The $\Delta CF_{R,I}$ in the scene description significantly deviated from that in free-viewing at the 6th fixation (p = 0.043, Cohen's d = 0.43, adjusted p-values (q-value) reported for FDR at $\alpha = 0.05$ for all 30 comparisons in Fig. 5c), while the free-viewing deviated significantly from the object counting condition at the 5th fixation ($p = 0.049$, Cohen's d = 0.93). No significant difference in $\Delta CF_{R,I}$ was observed between object counting and object search conditions. An analysis that used time-weighted fixations rather than fixation frequency resulted in similar findings (Figure Supplementary 3b, c).

## The role of gaze, head, and hand position in directing eye movements

Many of the images in our experiment present one or more individuals directing their gaze, head, or hands to objects that are typically relevant to the scene understanding. Thus, a possible explanation for our results is that observers' fixations on the SU-relevant locations and objects are a byproduct of the well-documented finding that observers often follow with their eyes the gaze[31,61-64] or anticipated action of others[65-67].

To show that the SU-relevant fixations are not solely a byproduct of following gaze, head, hand direction, or body posture but rather eye movements to objects critical to scene understanding, we conducted an experiment to assess which objects observers perceived to be grasped or gazed at for each scene. Figure 6a (left) details the experiment procedure. A separate group of participants (n=25) viewed the Winograd images with unlimited time and were asked to click the box they perceived to be grasped or gazed at by the person in the scene. Figure 6a (right) shows examples of scenes where a SU-relevant object was perceived by the majority of participants to be grasped or

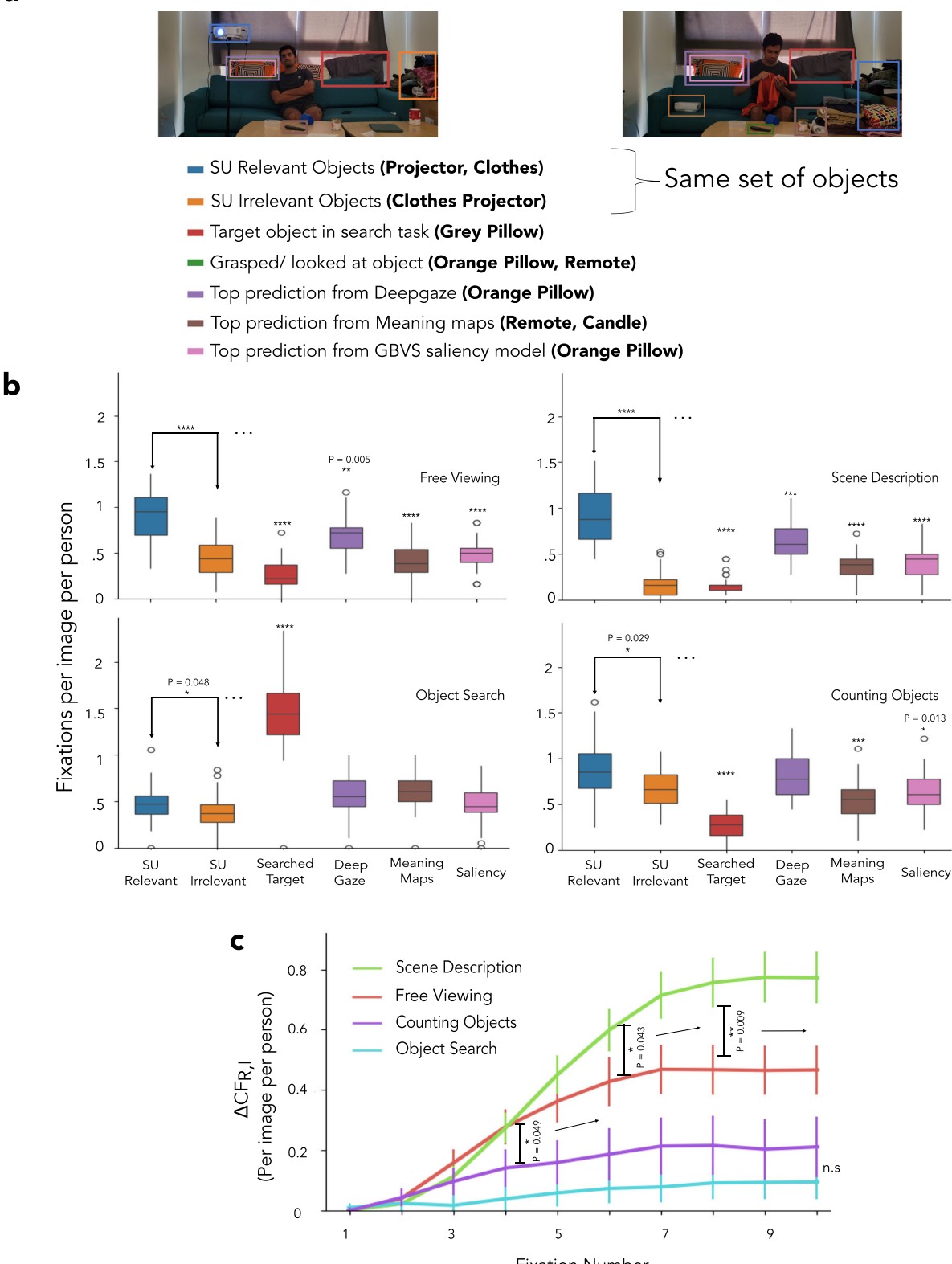

gazed at (top two images), and scenes where it was not (bottom two images). We found 20 images out of the 36 images for which the SU-relevant objects were not considered the object to be grasped or gazed at by the human in the image. We will refer to these scenes as the No Gaze image subset.

Analysis of the No Gaze image subset showed a similar pattern of results to our main findings (Figs. 2c, 4b, 5b). Observer fixation heat maps for free viewing were significantly more similar to the fixation heat maps from the scene description condition (FV-SD, $r = 0.5$) than to those from the object search (FV-OS, $r = 0.32$, $p < 0.001$, Cohen's d = 1.28) or counting objects (FV-CO, $r = 0.41$, $p = 0.030$, Cohen's d = 0.71) as shown in Fig. 6b.

The observer fixation heat map correlation across Winograd images (Winograd correlation, from seven pairs in the 20 images) for

**Fig. 5 | Fixation frequency for object categories. a** Bounding boxes for objects in each Winograd pair were manually created, and the object categories were assigned to them. The scene understanding relevant (SU-relevant) and the SU-irrelevant categories have the same set of objects across each Winograd pair. **b** Fixation frequencies based on the object categories in free viewing and scene description conditions across 25 participants per image (and 36 images) show that the SU-relevant object was most fixated ($p = 0.005$ for comparison of fixation frequency to SU-relevant and DeepGaze categories in free viewing; $p < 0.001$ for all other comparisons). The search target was most fixated in the object search condition ($p < 0.001$ for all comparisons). The SU-relevant object was fixated on the most for the counting objects, although the difference relative to the SU-irrelevant was smaller (p = 0.029). **c** $\Delta CF_{R,i}$: The difference in cumulative fixation frequency observed (for 25 participants per image, 36 images) between SU-relevant and SU-

irrelevant object categories was highest in the scene description condition (different from free viewing at 6th fixation onward, p = 0.043), followed by free viewing (different from counting objects at 4th fixation onward, p = 0.049). No significant difference was observed across any fixation for the other two conditions. (****$p < 0.0001$; ***$p < 0.001$; **$p < 0.01$; *$p < 0.05$). A one-tailed bootstrapped analysis was conducted to test the significance of all the results. All analyses had their significance levels corrected for the False Discovery Rate (28 comparisons, 20 shown here). In **b**, the box plot shows a participant-level distribution of fixation frequencies, and the line within the box indicates the median. The box spans the interquartile range (IQR), and the whiskers extend to the most extreme values within 1.5 × IQR. In **c**, the central measure is the mean, and the error bars show the 68% bootstrap confidence interval. Consent was obtained from the person in the featured image for its publication.

the free viewing condition was no different than the scene description condition ($p = 0.22$, Cohen's d = 0.56) but significantly lower than that observed for the object search ($p = 0.006$, Cohen's d = 1.34) and counting objects (p = 0.037, Cohen's d = 1.03) conditions (Fig. 6c). The Winograd correlation for the existing fixation prediction models (GBVS, meaning maps, and Deepgaze, Fig. 6c) was significantly higher than human correlations ($p < 0.001$ vs. free-viewing for all comparisons, Cohen's d = 1.91, 1.70, 1.23, respectively), while the correlation for the scene understanding map (SUM) was the lowest and not different from that observed in human fixation heat maps for the free viewing and scene description conditions (p > 0.9 in both conditions, Cohen's d = (0.20, 0.18) respectively).

In addition, similar to what we observe in the entire data set, for the No Gaze subset, observers also execute more frequent fixations on SU-relevant objects in the free viewing and scene description conditions ($p < 0.001$ for both conditions, Cohen's d = 1.26, 1.62, respectively) compared to the SU-irrelevant objects. However, unlike the entire data set, the No Gaze subset does not show a significant difference between the fixation frequencies to the SU-relevant and SU-irrelevant object categories for the object search (p = 0.282, Cohen's d = 0.14) and counting objects (p = 0.170, Cohen's d = 0.61) conditions, suggesting that for those two conditions, fixations on SU-relevant objects were driven mainly by gaze cues (images for which gaze cues point to the SU-relevant objects and which were not part of the No Gaze Subset). Finally, fixation frequencies for other object categories and the cumulative fixation frequency analysis show similar results for the No Gaze subset as in the entire image data set (see Figure Supplementary 4a, b).

To further assess whether observer eye movements are more guided by objects critical to scene understanding (SU-relevant objects) vs. objects perceived to be grasped/gazed at, we directly compared the frequency of fixations on the two categories. For the entire image dataset and the No Gaze image subset, SU-relevant objects were fixated more frequently than objects perceived to be grasped/gazed at (Fig. 6d, adjusted p-values (q-value) reported for FDR at $\alpha = 0.05$ for all 24 comparisons in the No Gaze image subset, 16 comparisons in Figure Supplementary 4a) in free-viewing ($p < 0.001$ for both image sets, Cohen's d = 1.19, 1.33, respectively), scene description ($p < 0.001$ for both image sets, Cohen's d = 0.92, 1.20, respectively) and counting objects conditions (p = 0.009, Cohen's d = 0.86 for the entire image set and p = 0.007, Cohen's d = 0.99 for the No Gaze subset), but not for the object search condition (p > 0.7 for both image sets, Cohen's d = 0.15, 0.20, respectively).

Together, all the analyses suggest that during free viewing or while describing scenes, observers most frequently direct their eyes not just to objects inferred to be grasped or gazed at but to objects critical to understanding the scene.

## Causal influence on scene understanding of fixating critical objects

More frequent fixations on SU-relevant objects during free viewing and scene description conditions might imply that processing those

objects with the high-resolution fovea is functionally important to maximize the understanding of the scene. We separately analyzed the scene descriptions condition trials for which observers correctly described the scenes and those for which they incorrectly described them (using a threshold based on cosine similarity of LLM embeddings of descriptions, see the methods section for details). Across all images, the difference in fixation frequency to SU-relevant vs. SU-irrelevant objects was significantly greater for trials that resulted in correct descriptions than for trials with incorrect descriptions ($p = 0.004$, Cohen's d = 0.53; see Figure Supplementary 5a for examples of descriptions near the threshold used for classifying a description to be correct/incorrect and see Figure Supplementary 5b, c for the overall results and variations of adopted threshold to classify descriptions as correct/incorrect). This association between eye movements and scene descriptions has been shown in previous studies.[46,47].

However, the association does not necessarily imply a causal influence of fixation on SU-relevant objects on accurate scene understanding. An alternative explanation is that observers process the entire scene and extract its meaning before eye movements. In this alternative explanation, observer fixations on SU-relevant objects are a subsequent process of looking at objects critical to the already extracted scene understanding. In this interpretation, fixations on the SU-relevant objects are not required to understand the scenes accurately.

We conducted a separate experiment to evaluate the causal influence of fixations on SU-relevant objects on scene understanding. Two new groups of 15 observers maintained fixation either at the SU-relevant or the SU-irrelevant objects during a 500 ms presentation of the same test images used in experiment 1. The observers then described what was happening in the scene. The collected descriptions were compared to the gold standard description (descriptions from observers with unlimited time and eye movements) using sentence similarity measures with the embeddings of Large Language Models (LLMs; see methods for validation of Gemini similarity scores with human ratings). Figure 7a, b shows the procedural workflow of the experiment and an example description from observers fixating on the SU-relevant or SU-irrelevant object for a sample image. Figure 7c, d shows that observers' descriptions were more similar to the gold standard description of the scene when they fixated on SU-relevant objects compared to when they fixated on SU-irrelevant objects ($p < 0.001$, Cohen's d = 1.18). For reference, Fig. 7d shows an upper bound of similarity of descriptions provided by the average score across different gold standard descriptions (different observers describing the scene while exploring the image with unlimited time). A lower bound on similarity scores was estimated by permuting scene descriptions across images and comparing their similarity to the gold standard from other unmatched images (see methods. Also, see Figure Supplementary 5d for similar results obtained using embeddings from the GPT4 LLM model rather than Gemini).

Together, the results suggest that fixating on SU-relevant objects during free viewing causally leads to more accurate descriptions of the scenes than when fixating on SU-irrelevant objects.

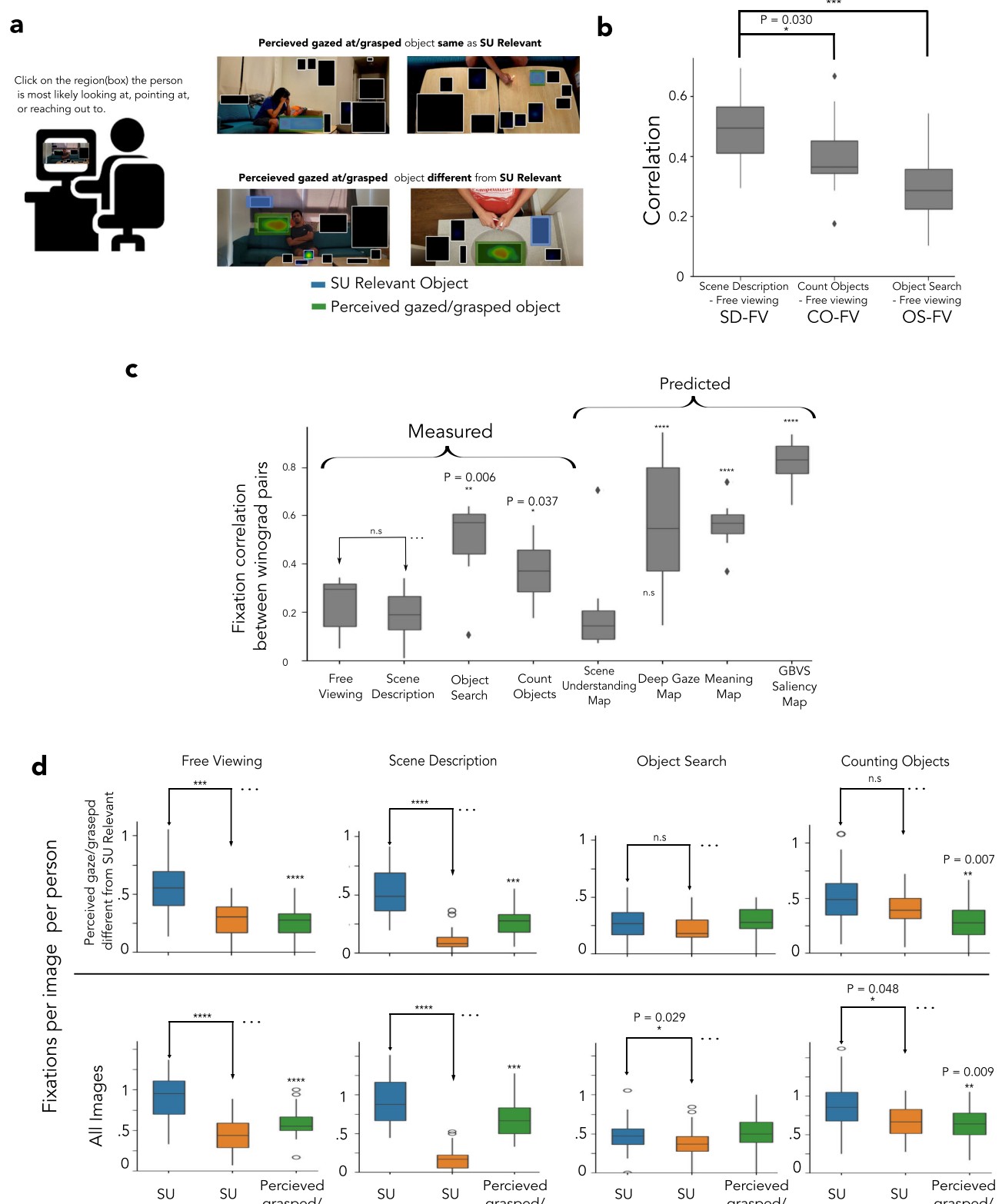

## Fixations on people and their role in the understanding of scenes

All presented analyses involved fixations on objects in the scene and excluded the fixations on people (hands, faces, bodies) to isolate the influence on fixations of SU-relevant objects, which varied across Winograd images (while the people in the scene were present in both). However, the larger literature documenting the importance of fixations on people's heads[49,50,68] and hands[69] motivated us to understand how such fixations fit within the theoretical framework of eye movements to maximize scene understanding.

We first re-estimated the SUMs by collecting descriptions of the scenes that included the deletion of people or people's hands in the scenes (in addition to the deletion of objects). Figure 8a presents a sample SUM showing that the person's hands in the scenes were the

**Fig. 6 | Analysis of the No gaze image subset. a** Procedure to determine objects (left) that were perceived to be grasped/gazed at. Participants were asked to click on box locations they perceived to be grasped/gazed at by the person in each scene. Examples of images where the participants perceived the grasped/gazed-at object to be the same as the SU-relevant object (right, top 2) and images where it was different than the SU-relevant object (right, bottom 2, No gaze image subset). The top left icon is from Flaticon.com (icon ID 47865). **b** Fixation heat map correlations across conditions (25 participants per image, 20 images) for the No gaze image subset were similar to the pattern of correlations for the entire image set (p = 0.03 for SD-FV vs CO-FV; *p* < 0.001 for SD-FV vs OS-FV) (Fig. 2c); **c** Fixation heat map correlations between Winograd image pairs (25 participants per image, 7 image pairs) for the different conditions of the No gaze image set were similar to the entire image set (*p* = 0.037 vs CO and p = 0.006 for OS, *p* < 0.001 for all other comparisons

with FV; Fig. 4b). **d** Fixation frequencies for object categories for the No Gaze image subset (25 participants per image, 20 images) show that the SU-relevant category is fixated more frequently than objects perceived to be grasped/gazed at (top row) in SD and FV conditions (*p* < 0.001 for all comparisons). This result holds even when all images were included (bottom row). However, the SU-relevant and SU-irrelevant object difference was not significant for OS (*p* = 0.282) and CO (*p* = 0.170) conditions in the No Gaze image subset. (****\*p* < 0.0001; ***\*p* < 0.001; **\*p* < 0.01; *\*p* < 0.05 from one-tailed bootstrap). Comparisons in **d** No Gaze subset used adjusted p-values with False Discovery Rate (*n* = 24, some in Figure Supplementary 4a). Box, IQR, and whiskers as defined in other figures. IQR across images for **b** and **c** and across participants for **d**. Consent was obtained from the person in the featured image for its publication.

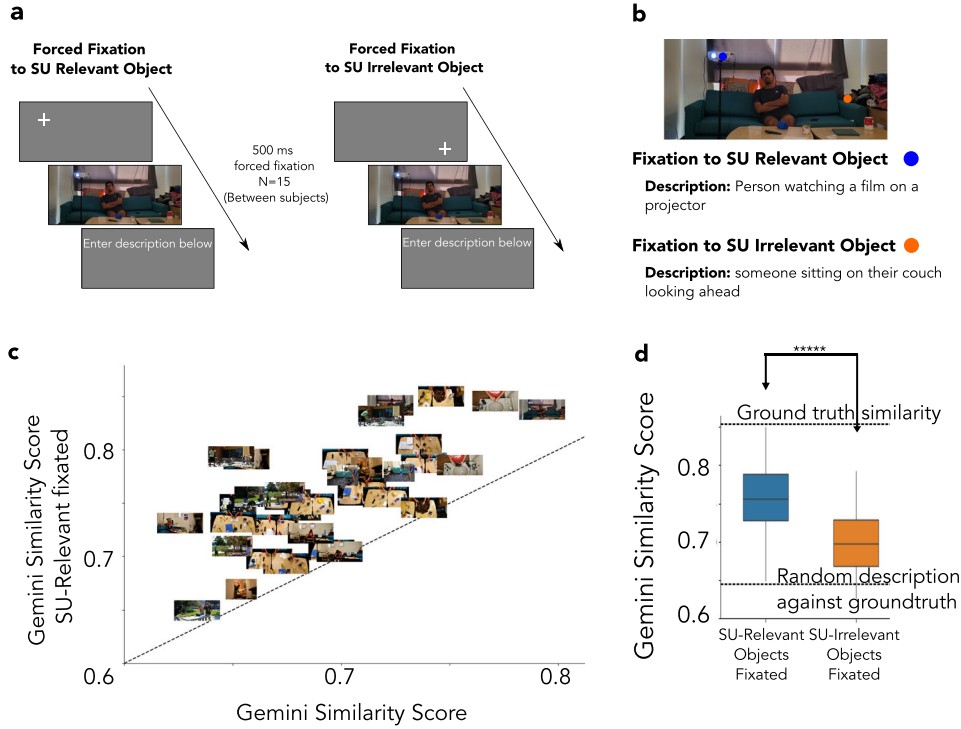

**Fig. 7 | Forced fixation at SU-relevant and SU-irrelevant locations. a** A between-subjects (N=15) experimental design was used for the forced fixation experiment. Participants were instructed to maintain fixation on SU-relevant or SU-irrelevant objects for 500 ms, while real-time eye-tracking ensured fixation was maintained. They were asked to provide a best-guess description of what was happening in the scene. **b** Examples show that observers' descriptions were more similar to the gold standard meaning of a scene when they fixated on the SU-relevant objects. **c** The scatter plot shows the similarity scores (Gemini, see Figure Supplementary 1b, c for comparison of Gemini similarity scores and human ratings) of the scene descriptions relative to the gold standard for each image (averaged across observers) while fixating SU-relevant objects vs. while fixating SU-irrelevant objects. **d** Average similarity score relative to the gold standard description for the descriptions provided by participants fixating on either SU-relevant or SU-irrelevant (15 participants

per fixation location per image, 36 images). The similarity of the descriptions was significantly higher (*p* < 0.001) when participants were fixating at the SU-relevant locations compared to the SU-irrelevant locations. The top dashed line indicates an upper bound established by the average similarity score across gold standards (inter-observer agreement of descriptions with unlimited time). The bottom dashed line indicates a lower bound calculated by permuting descriptions across images and comparing them to the gold standard from unmatched images. **\*\*\*\*** (*p* < 0.00001). A one-tailed bootstrapped analysis was conducted to test the significance of all the results. In **d**, the line within the box indicates the median. The box spans the interquartile range (IQR), and the whiskers extend to the most extreme values within 1.5 × IQR across images. Consent was obtained from the person in the featured image for its publication.

most critical (altered the scene description the most). We found that the people were the most critical in 77.78% (28 images) of the images in the dataset, while the SU-relevant object was in the remaining.

We then repeated all analyses from previous sections, but included the fixations on people in the scenes. As with the initial analyses excluding fixations on people, we found higher correlations of observer fixation heat maps of the free viewing with those of the scene description condition (FV-SD, r = 0.66) than with the fixation heat maps of counting objects and object search conditions (Fig. 8b; *p* < 0.001 in both comparisons, Cohen's d = 1.50, 1.70, respectively). The presence of fixations on people increased the correlations across

fixation heat maps of the Winograd image pairs for all conditions and images. Although the results remain the same as the analysis excluding fixations on people (see Figure Supplementary 6a), the fixation heat map correlation across the Winograd image pairs for the counting objects condition was no longer significantly different from that for the free viewing condition (p = 0.26, Cohen's d = 0.45).

We also re-analyzed the eye movement data for the four conditions to assess the fixation frequency on people in the scenes (their hands, bodies, or faces; see methods). Observers fixated significantly more frequently on people for the scene description condition (Fig. 8c; *p* < 0.001, Cohen's d = 1.45). Fixations on people in free viewing were

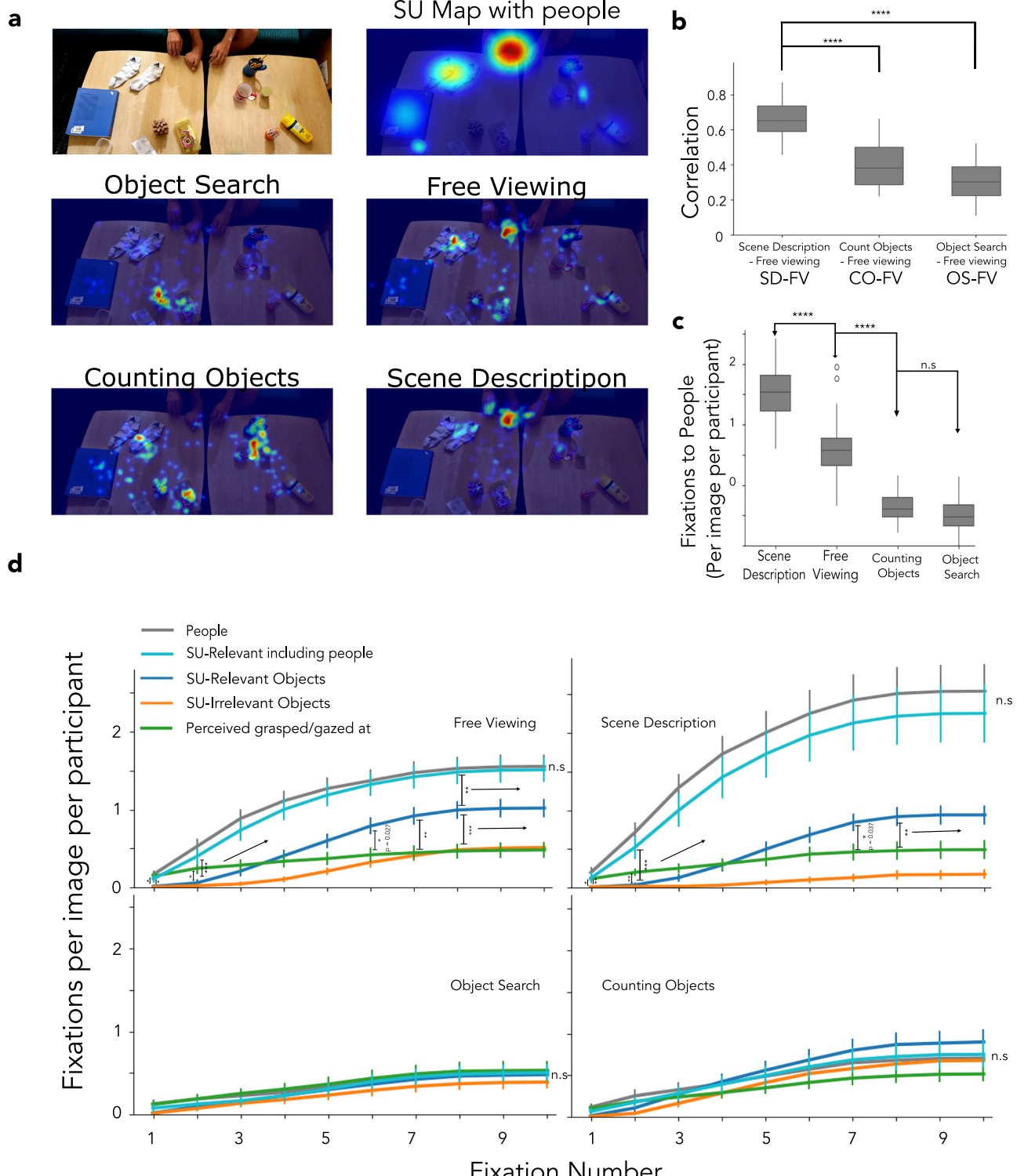

significantly more frequent than those observed in the object counting and search conditions ($p < 0.001$ for both comparisons, Cohen's d = 1.59, 1.65, respectively).

We calculated the cumulative fixation frequencies to understand the temporal dynamics of fixation selection on different scene components (people, SU-relevant objects, perceived to be grasped or gazed at objects, and SU-irrelevant objects). We used the No-Gaze image subset for which the perceived grasped/gazed-at objects were distinct from the SU-relevant objects (see Figure Supplementary 6b for similar results for the set of images in our dataset where the grasped/

gazed at objects were the same as the SU-relevant objects). Figure 8d shows that starting with the 1st fixation, observers most frequently fixated on people (or the highest value of the scene understanding map estimated with the inclusion of people, see SU-relevant including people in Fig. 8d, adjusted p-values (q-value) reported for FDR at $\alpha = 0.05$ for all 30 comparisons) for free viewing and scene description. For these conditions, early fixations (1st and 2nd) were also more frequently directed to objects perceived to be grasped or gazed at significantly above the SU-relevant object category (free viewing: $p = 0.011$, Cohen's d = 1.3 for the 1st fixation; $p = 0.016$, Cohen's d = 1.13 for

**Fig. 8 | Recomputing Scene Understanding Maps with people. a** Scene understanding map (top right) when the removal of people from the scenes was included in the procedure. The SUM predicts people as the most fixated, followed by the SU-relevant object. Similar fixations were observed in both free-viewing (FV) and scene description (SD) conditions, which differed from those in object search (OS) or counting objects (CO) conditions (25 participants per image, 36 images). **b** After including people in the scenes, fixation heat maps for the FV condition correlated the most with the SD condition ($p < 0.001$ for all comparisons). **c** Fixation frequency on people in the scenes was highest in the SD condition ($p < 0.001$ vs. FV), followed by FV ($p < 0.001$ vs. CO). **d** The cumulative fixation for the No Gaze image subset (20 images) shows significantly higher fixation frequency on people than on SU-relevant objects for the FV ($p = 0.021$ for the first fixation) and SD ($p = 0.009$ for the first fixation) conditions, starting with the first fixation onward. The SUM with

people also predicts a higher fixation frequency on people. Initially, observers fixated on the to-be-grasped/gazed at objects significantly more than the SU-relevant objects for the FV ($p = 0.011$, $p = 0.016$ for the first two fixations, respectively) and SD ($p = 0.011$, $p = 0.003$ for the first two fixations, respectively) conditions. However, the cumulative fixations on SU-relevant objects become significantly higher than those on the to-be-grasped/gazed at objects from the 6th (FV, $p = 0.027$) or 7th fixation (SD, $p = 0.037$) onward. No significant differences were observed across any fixation for OS and CO. (****$p < 0.0001$; ***$p < 0.001$; **$p < 0.01$; *$p < 0.05$; one-tailed bootstrap). Comparisons in **d** used adjusted p-values using False Discovery Rate (n=30). In **b** and **c**, the line within the box indicates the median. Box, IQR, and whiskers as defined in other figures. In **d**, the central measure is the mean, and error bars show the 68% bootstrap confidence. Consent was obtained from the person in the featured image for its publication.

the second fixation; scene description: $p = 0.011$, Cohen's d = 1.26 for the first fixation, $p = 0.003$, Cohen's d = 1.23 for the second fixation). Subsequently, the cumulative fixations on SU-relevant objects were significantly higher than the perceived grasped/gazed-at from the 6th fixation onward for free-viewing ($p = 0.027$ at the 6th fixation, Cohen's d = 0.95) and the 7th fixation onward for scene description ($p = 0.039$ at the 7th fixation, Cohen's d = 0.97) condition. We observed no statistically significant differences in cumulative fixations across scene components for the object counting and search conditions.

Similar to the AUROC analysis without people, AUROC analysis including fixations on people also resulted in a significant difference between SUM and GBVS saliency maps ($p < 0.001$, Cohen's d = 1.87) and no statistically significant difference for DeepGaze, meaning maps, and SUMs, though SUMs were lower than meaning maps and DeepGaze ($p > 0.09$ for both comparisons; see Figure Supplementary 6c).

## Discussion

### Eye movements during free viewing are directed to objects critical to understanding scenes

The purpose of our study was to gain an understanding of the image properties that guide eye movements during free viewing of scenes. Although there has been a long-held view that low-level saliency drives eye movements during free viewing[36,37,40–42], recent studies have supported the concept that eye movements are directed to regions/patches of a scene judged to be meaningful[44,45,70]. Here, we explored the hypothesis that humans aim to maximize scene understanding accuracy during free viewing and move their eyes to elements (people and objects) critical to scene understanding. The most locally meaningful region, object, or person (meaning map maximum) is not necessarily the most critical person/object for understanding a scene. The latter involves relationships between people, objects, and regions that contribute to the overall understanding of the scene, which is not present in meaning maps.

We first investigated human eye movements to objects for different instruction conditions with newly designed stimuli (Winograd images). The Winograd image pairs varied the objects critical to scene understanding while maintaining visual saliency and the meaningfulness of local regions. Three findings support our hypothesis. First, fixation distributions during free viewing were most similar to those of observers when describing the scenes. Their correlation was comparable to the inter-observer fixation distribution agreement for the free viewing condition and similar to previously reported correlations between the best models and human fixations (e.g., GBVS-human fixation correlation = 0.45,[71]). Second, small visual alterations to the images that changed the meaning of the entire scene (Winograd images[60]) influenced human fixations during free viewing. Third, fixations were more frequently directed to the objects critical to understanding the scene rather than the most salient or locally meaningful region.

Previous studies have shown how scene gist, object spatial relations, and object function relations guide eye movements when

observers are searching for objects[28,72–81]. However, this had not been demonstrated for free viewing due to the difficulty of determining what components of the semantic information present in the scene were relevant to free viewing eye movement guidance and how to quantify these semantic components. Here, we developed a method to quantify the semantic role of an object in free viewing by measuring how the object's removal alters scene understanding (scene description). This is distinct from recent efforts to quantify the semantic relationship among objects in scenes (concept maps[82]), which do not define their relationship to the scene's understanding. Such concept maps do not predict the object observers fixate the most nor the influence of the Winograd images on human fixations in our study (see Figure Supplementary 7a–c).

### Eye movements to objects critical to understanding scenes vs. objects perceived to be grasped/gazed at

Some of our images contained a person directing their gaze or head toward the object critical to the scene understanding or the presence of hands, which could suggest an interpretation that observer eye movements are simply showing the well-documented behavior of following the gaze of other people in the scenes or fixating on objects perceived to-be-grasped/gazed-at[31,61–67].

However, we analyzed images with no grasp/gaze cues and directly compared fixations on objects judged to be grasped/gazed vs. objects critical to scene understanding. Our findings show that objects critical to scene understanding are frequently looked at even when the gaze is judged to be directed elsewhere or the to-be-grasped/gazed-at object is other than SU-relevant objects. However, gaze, head, and body cues are important predictors of the spatial location of these critical objects, and observers initially fixate on perceived-to-be-grasped or gazed at objects, consistent with gaze cueing studies[83,84]. However, even without the grasp/gaze cues, the observer's eye movements eventually fixate on the SU-relevant objects. We argue that objects critical to scene understanding are a general concept guiding eye movement fixations during free viewing and that the gaze of others, most often, but not always, serves as a predictive cue of the location of critical objects.

### Functional role of fixations to maximize scene understanding

We found that trials in which a person did not fixate on the critical object were associated with less accurate scene descriptions. A follow-up experiment that manipulated the point of fixation in the scene evaluated the causal influence of fixation on extracting the information required to understand a scene. We demonstrated that fixating on the object critical to scene understanding yielded more accurate scene descriptions than fixating on a different, irrelevant object. This suggests that observers' fixations on objects during free viewing have a functional purpose to maximize scene understanding. This is consistent with many recent findings demonstrating the functional perceptual role of saccades[17,19,32,64,85], micro-saccades[86], and smooth pursuit eye movements[14,87].

Our functional evaluation only evaluated a fixation on the object critical to the scene understanding, and another fixation on another object at a comparable distance from the center of the image. We did not evaluate an array of fixations that covered the entire image. It might not always be necessary to fixate on the center of an object. Fixating close to the critical object (within 2 degrees) may also result in maximal accuracy in describing the scenes. And for some scenes, it may be that center-of-mass fixations between the critical object and the hands might also provide accurate scene description[32,88,89].

The findings should not be interpreted to suggest that observers do not use peripheral information to guide their eye movements. For example, observers might identify a tool in the visual periphery but might not be able to specify the tool type. The peripheral information can guide the eye movement to the tool, and only after fixating on the object can the observer identify it as a screwdriver and more accurately describe the scene.

The results do not suggest that a single fixation is sufficient to understand a scene. As with other tasks with scenes[90–94], many of our scenes might require multiple fixations to extract their meaning accurately. The planning of multiple fixations might also consider the motor cost of saccades and result in fixating irrelevant objects placed between two objects critical to scene understanding[95,96].

### The role of fixations on people and social cues in the theoretical framework of scene understanding

Humans looking at other people in scenes has been a universal finding across the eye movement literature[2,49,50,68,69]. Our results follow the classic finding. During free viewing and scene description, people in the scenes were fixated the most, well above any objects, including those critical to scene understanding. These findings are consistent with the theoretical framework of eye movements aimed at maximizing scene understanding. Erasing people from the scenes had the largest impact on the scene description when we re-created the scene understanding maps (SUMs) to assess the influence of people.

### Comparing scene understanding maps to saliency models, DeepGaze, and meaning maps

A saliency model based on low-level image features[4] failed to predict the high frequency of human fixations on objects critical to understanding the scene. This is because low-level saliency models do not incorporate any aspects of the object relationships or tasks. Meaning maps[44] also fall short for our images because they consider the local meaningfulness of objects and not their contributions to understanding the entire scene. DeepGaze[59] is trained on human fixations and image features and thus, in principle, should be able to learn to predict fixations on the objects critical to scene understanding. DeepGaze's lower prediction of frequent human fixations for our images may reflect the data used to train DeepGaze. The training data set might not include sufficient samples of images with complex behaviors as depicted in our Winograd images. Additionally, DeepGaze's emphasis on fixation density maps may make it less capable of accurately predicting the most frequently fixated object. In addition, SUM's advantage over the other models is not present when considering all human fixation density maps. SUMs were marginally better (though not significantly different) than DeepGaze and meaning maps when considering only fixations on objects, and were marginally lower (though not significantly different) than the two models when including fixations on people.

Importantly, even if DeepGaze predicted each scene's most fixated object, it would not provide a unified theoretical understanding of what drives certain objects to be fixated the most for each image. Our experimental investigation provides such a theoretical account. It explains free viewing and frequent fixations on people and objects in terms of their contributions to understanding the scene.

### Why do people seek to understand scenes during free viewing?

Our findings suggest that when observers view scenes without a specific task instructed by the investigator (free viewing), seeking visual information to understand the scene is one of the default tasks they engage in. Understanding a scene is essential for contextualizing the current visual input, making inferences about likely past events that have led to the current visual state, and, importantly, predicting future events[97–100]. Making inferences requires an observer to consider all likely possibilities for that scene. In this framework, eye movements during free viewing of scenes are directed to locations that reduce the uncertainty of the possible states of the visual world[101,102], consistent with concepts of information sampling and curiosity[103].

Our findings also show a smaller but significant effect of fixating on objects critical to the scene's understanding, even for the object-counting and search conditions. Our analysis reveals that these effects were primarily driven by grasp/gaze cues.

Consistent with our findings suggesting that understanding a scene is one of the default observer tasks during free viewing, a recent fMRI study showed that the semantic information of a scene (represented by the embeddings of a deep neural network visual-language model) predicted brain activity of people passively (no task) viewing scenes better than traditional object labels[104].

### Generalization to other image types and influence of individual and cultural differences

All scenes in the Winograd pairs feature one or more individuals or a person's hand, implying an action or social interaction. These types of images are fundamental to human daily lives. However, understanding these scenes can often change due to the presence or absence of a single object, thus requiring observers to explore the image with their eye movements. How would the analyses apply to images that contain objects spread across an image (e.g., a prototypical kitchen or forest image) with no implied future or past actions or social interactions? For such simpler images, there might not be a single object that is critical to their understanding. Furthermore, observers might be able to rapidly understand the scene with peripheral visual processing and without eye movement exploration[105–107]. Thus, we might expect fixations with those images to have a lower perceptual function and the most frequently fixated object to be less related to its relevance to scene understanding. Additionally, unlike the Winograd image dataset, many real-world scenes feature multiple people, some of whom are critical to understanding the scene, while others are not. It is likely that the SUMs, but not the meaning maps, best predict which person or people in the scenes are fixated the most, but this will need to be tested.

In addition, a limitation of our approach is that it might not capture individual[108] and cross-cultural differences[109] in scene understanding that might influence the most fixated object in scenes. Individual and culture-specific SUMs may be a possible direction for predicting such inter-individual and inter-cultural differences in fixation behaviors.

To conclude, our findings suggest that during free viewing of scenes, eye movements are directed to people and objects that are critical to understanding the scene, rather than to regions that are visually salient or judged to be meaningful. The eye movement fixations serve a perceptual function, as they causally improve the accuracy of scene descriptions. The theory of eye movements to maximize scene understanding, along with its empirical implementation through scene understanding maps, provides a unified account of free viewing of scenes. They predict which objects are fixated on frequently, explain the higher frequency of fixations on people over objects, and frequent fixations on objects that are perceived to be grasped or gazed at. Together, our findings suggest that an important default task for the human brain during free viewing is comprehending the visual world.

## Methods

### Winograd image pairs (WI)

The experiment stimuli include 20 pairs of Winograd images photographed within the University of California, Santa Barbara premises. The images include indoor, outdoor, and table scenes. Each pair was ensured to have almost the same set of objects and positions. The small visual changes across image pairs could be manipulations of an object's position, its substitution by another object, or an actor's posture. The pairs were split into two sets: Winograd Set1 and Winograd Set2[60]. The changes greatly alter how the scene was described while aiming to preserve the lower-level saliency and meaning maps. However, some changes would be introduced in the saliency and/or meaning maps by replacing an object or a variation of its location.

Each image pair was carefully curated to ensure that at least five random people described the pair differently and were consistent for a given image. Two pairs of Winograd images were removed from the dataset because their gold standard descriptions were inconsistent and vague. Out of 180 descriptions (from the other 36 images * 5 descriptions each) in our entire dataset, 149 were consistent with their corresponding image (at least three descriptions were consistent for each image). All of these images feature a person (or parts of a body) and tend to depict some future action or behavior that forms the scene's understanding.

### Participant information and informed consent

For all eye tracking experiments, the participants (260 participants in total) were undergraduate and graduate students of UCSB Psychological Brain Sciences. Participants did not know a priori about the hypothesis or the details of the experiment. Although the main study was conducted with participants ranging from 18 to 30 years, we expect the main findings to generalize to other ages. We collected self-reported ethnicity and gender data, but due to a technical issue, we have the data for only 100 out of 260 observers. We had consent to collect the gender and ethnicity data. We also collected Amazon Mechanical Turk data from US workers (276 participants in total) for our online studies, but we did not collect ethnicity or gender data from those participants. Race, gender, and ethnicity were not considered in the study design. Our main hypothesis is that eyes are directed to regions/objects that maximize scene understanding and are presumed to apply to people of all genders and ethnicities.

Participants provided written consent before participating in all experiments. This was approved by the Office of Research and Human Subjects, University of California, Santa Barbara. The authors affirm that all participants who took part as actors in the Winograd scenes provided informed consent for the publication of all images in the dataset.

### Eye tracking and experimental setup

Eye movements were recorded using an EyeLink 1000+ desktop-mount eye tracker (spatial resolution: 0.01°) with a sampling rate of 500 Hz. Participants sat 75 cm away from a 19-inch monitor, so that the screen subtended a visual angle of 26. 6° × 21. 8° at 1,280 × 1024 pixels. The height of a stimulus image was approximately 12.7° of visual angle. Head movements were minimized using a chin and forehead rest. A velocity threshold of 22°/sec and an acceleration threshold of 4000°/sec$^2$ were used for the detection of saccades. Eye movements were recorded from the left eye. The experiment was controlled with SR Research Experiment Builder software.

### Eye movement experiment with four conditions

In the first experiment, participants (between subject design) viewed the images in four conditions: free viewing, scene description, object search, and counting objects. Each condition involved 50 participants who were asked to complete 18 trials in which they viewed one set of Winograd images from all pairs (25 participants per image). Each image

was shown for two seconds and was followed by condition-specific instructions. Below, we describe the instructions given to participants for the four conditions.

**Free viewing.** Participants were instructed to view the scene naturally. No explicit tasks or instructions were given to them.

**Scene description.** Participants were instructed to describe the presented scenes. After they viewed the scene, they typed their description with no time limit.

**Object search.** Participants were instructed to search for an object within the scene. Before the start of the trial, they were presented with the name of the object they needed to search for. After the trial, they reported back whether the object was on the left or right side of the image. By design, the image contained the object searched at the same location across the Winograd image pair.

**Counting objects.** Participants were instructed to count the objects on each image's left and right sides. After the trial, they reported which side of the image had more objects (see Fig. 1b for the experiment's procedural flow diagram.

### Fixation prediction models

The study used four fixation prediction models. The models' fixation prediction heat maps were compared against the fixation heat maps collectively made by participants in each of our eye-tracking experiments.

**Graph based visual saliency (GBVS).** A bottom-up saliency model proposed by Harel et al.[4] constructs a computational graph using Markov chains on top of the extracted image features to generate a heat map of possible fixation locations. Our study uses an implementation provided by Kümmerer[110] for the GBVS model. We chose the GBVS model because it is one of the top saliency models[111] relying purely on low-level image features to compute its saliency map, true to the original definition of saliency.

**DeepGaze.** DeepGaze is a neural network model trained on image features extracted from VGG-19 Convolutional Neural Network (CNN)[112] along with human fixations on those images while free viewing[59]. The model produces a heat map of possible fixation locations given an image. This study uses an implementation provided by Kümmerer[113].

**Meaning maps.** Meaning maps is a crowd-sourced model developed by Henderson et al.[44], which uses the subjective ratings provided by people on how meaningful they find local circular patches of an image. Each image was divided into overlapping circular patches, which were then randomized across all images. Different individuals then rated small portions of these patches to determine their meaningfulness. The procedure ensured each patch of images had three raters, and the overlap between the patches ensured that any region of the image had ratings from 27 raters. To facilitate the creation of meaning maps for all the images in our study, 48 Amazon Mechanical Turk participants rated approximately 300 circular patches, each measuring 3 dva in size. The final result was a map of meaningful locations in a scene as predictors of locations people might fixate on while viewing the scenes. The procedure proposed by Henderson et al.[44] also included creating and rating 7 dva patches to compose a 7 dva meaning map and averaging it with the 3 dva meaning map. We collected 7 dva meaning maps but did not use them in our presented meaning maps. Since our study image height was 12.7°, the 7 dva patch resulted in every patch having multiple objects that could be recognized by the raters, and making most of the image regions meaningful. Thus, the 7 dva

meaning maps were uniformly meaningful, and averaging them with the 3 dva maps did not change the resulting meaning maps. Thus, our presented results show the 3 dva maps, but including the 7 dva maps did not alter the results. Refer to Fig. 3b for a procedural flow chart of all these models.

### Scene understanding map

The scene understanding map visualizes an object's contributions to understanding a scene by quantifying the change measured in participants' descriptions of the scene after the object was removed. Creating a scene understanding map requires several steps described below: 1) Creating images with individual objects removed; 2) Collecting descriptions of images with objects removed; 3) Establishing a gold standard description for the intact image; 4) Assessing the similarity of the scene descriptions with an object removed to the scene description of the intact image; 5) Generating the scene understanding map. Below, we describe each of these steps.

**Creating images with individual objects removed.** Each image of the 18 Winograd pairs (36 images) was digitally manipulated to create versions with one object removed at a time. The resulting total number of images was 330. Each image had a total of 5 to 10 objects removed. Figure 3a shows an example of an image with some of its digitally manipulated versions. The photo editor app on the Samsung Galaxy S21 (version 3.4.2.43) was used to remove objects from these scenes.

**Descriptions of images with objects removed.** The dataset was split across the Winograd set, so participants viewed only one of the images of each Winograd pair. To ensure that participants do not see more than one version of each image, each set was divided into 11 unequal groups, each containing exactly one copy (original or manipulated) of each image. This implied that all participants provided descriptions for at most 18 images. One hundred and ten Amazon Mechanical Turk participants took part in this study. We collected five descriptions for each image (or its digitally manipulated copies) in the dataset. Before the online experiment, five images were used as a pre-test to ensure that observers followed the task instructions. Observers who did not accurately describe the pre-test images were not allowed to proceed to the main test with the 18 images.

**Gold standard descriptions.** To identify the impact of an object on scene understanding, we compared the descriptions of the original image to those collected for the manipulated image that had that specific object removed. We collected five descriptions for each of the original Winograd images. The gold standard description was defined as the best description among the five for each image. To establish which description was best for each image, a new group of fifty Amazon Mechanical Turk participants (twenty-five per Winograd Image set) selected the description that best described each of the original 18 Winograd pairs. Participants were randomly assigned to one of the Winograd pair sets (18 images). In each trial, the participant was shown an image with 7 descriptions (5 descriptions given to the image + two random descriptions from other images). Participants had to select the description that best described the presented scene. The two random descriptions were used to identify participants who did not follow the instructions. All data from participants who selected random descriptions were eliminated from the study. Importantly, the description with the highest vote (as the best description) was chosen for each image as the gold standard description.

**Description similarity ratings to determine the contribution of objects to scene description.** Eighteen Amazon Mechanical Turk participants were asked to rate the similarity of the descriptions using a scale from 1 to 10, with 10 indicating highly similar and 1 indicating very low similarity. In each trial, the participant saw the gold standard description for an image, followed by descriptions corresponding to each object being removed from the scene. Since there were five such description sets for each Winograd image, each participant had to finish 180 trials (36 images * 5 sets of descriptions). Refer to Fig. 3a for the procedural flow chart of this process.

**Heat map generation for scene understanding map.** To generate a heat map that visualizes the contribution of each object present in the scene to the scene description, we inverted the scale to have higher scores for dissimilar descriptions. Then we computed the median rating score corresponding to the description when each object was removed. The median score was assigned to all the pixels within the object's corresponding bounding box (defined in methods; refer to Fig. 3a). To highlight relative contributions within each image, we normalized the ratings by subtracting the lowest-rated object's rating from all the other objects' ratings and then by dividing the map by the highest score for each image (scale-inverted normalized ratings). We followed Stoll et al.[43] to generate the preferred fixation location in the images by using scale-inverted, normalized ratings corresponding to the removal of each object from the scene. Using the bounding boxes for our object, we modeled the ratings as 2D Gaussians centered at each object, with its amplitude corresponding to scale-inverted normalized rating and the Gaussian horizontal and vertical standard deviation derived as a fraction of the size of the bounding box (fraction=0.29 along the x-direction and fraction=0.34 along the y-direction). Figure 3a shows an example of a scene understanding map.

**Defining SU-relevant and SU-irrelevant objects.** Objects in each scene were categorized based on their scene understanding map score (SU-relevant and SU-irrelevant). Objects that belong to the SU-relevant category had the highest impact (most critical to scene understanding among objects in that image) on participants' scene descriptions when erased. If erasing them did not impact the participants' scene description, they would belong to the SU-irrelevant category. For each Winograd pair, the same set of objects belonged to the SU-relevant category in one image and to the SU-irrelevant category in the other image of the pair (refer to Fig. 3a). Our analyses focused on objects that were SU-relevant for one Winograd image and SU-irrelevant for the other. We did not focus on objects that had a low impact on the participants' scene descriptions for both images of a Winograd pair.

### Measuring objects to be grasped and/or gazed at in the dataset

Fifty Amazon Mechanical Turk participants were divided equally between the Winograd sets and were shown the images with objects in the scene covered with black boxes. This was done to isolate the perceived grasped or gazed at object from any contextual information that could influence participants' judgments. Participants were asked to click on the box they perceived to be grasped or gazed at by the person in the scene. The box with the maximum number of selections was considered the object perceived to be grasped or gazed at for each image. Figure 6a shows the experimental procedure and some examples. Images, where the SU-relevant object differed from the object perceived to be grasped/gazed at, constitute the No Gaze image subset.

### Forced fixation scene description experiment

Sixty participants were split into four equal groups, and each group saw 18 trials. Two groups were assigned to images from Winograd set 1, while the others were assigned to set 2. A between-subjects design across the Winograd pairs ensured that observers did not use their knowledge of one image to interpret its corresponding pair. Within each set, the two groups were asked to fixate on nine SU-relevant and nine SU-irrelevant locations. Each group saw a unique combination of image and fixation location. The images were presented for 500 ms. All trials where eye movements (saccades larger than one dva) were

detected within the presentation interval were discarded. On average, the discarded trials accounted for thirteen percent of all trials, resulting in thirteen descriptions per image and fixation location. Participants described each scene after it was presented. Because of the atypical nature of the task, observers were instructed to provide their best guess of what was happening in the scene.

### Repeating analyses including fixations/predictions on people in the scenes

We repeated the procedure to generate scene understanding maps (SUMs) but incorporated the influence of people in the scenes. Similarly, we also included the fixation predictions of the other models on people. We generated the measured fixation heat maps, including fixations on people. With these updated fixation/prediction heat maps, we recomputed the correlations of fixation heat maps across conditions and within the Winograd pairs for each condition. We also added fixations on people and the prediction of SU-relevant with the people category in the cumulative fixation distribution plots. Finally, we performed the AUROC analysis for all fixation prediction models, including fixations/predictions on people.

### Dependent variables

**Heat map generation for measured fixations.** Fixations on an image (a pixel white dot on a blank canvas) were convolved with a Gaussian kernel of standard deviation 0.5° visual angle. OpenCV2's GaussianBlur function was used for this purpose. These convolved fixation points were added onto a blank map using their x and y locations to generate the heat map.

**Selecting top predictions from fixation models.** Fixation prediction models can sometimes predict contrastive regions that are not objects of importance. These regions were identified by finding the maximum locations determined by the convolution between the map and a 2 dva uniform and normalized circular patch. If the model's prediction landed on an object, the object's bounding box was used to represent its prediction. If not, a bounding box was generated with the average dimensions of all the bounding boxes in the image.

**AUROC analysis for the fixation prediction models.** The shuffled AUC (sAUC) technique[114] was used for ROC analysis to evaluate the performance of prediction models in predicting human fixation heat maps. For each image and observer combination, the prediction maps were thresholded at different levels, and the true positives (TPs) at each level were calculated by counting the number of fixations that fell in the areas above the thresholded level. The false positives (FPs) at each threshold level were calculated by sampling fixations (the same number of fixations as the positive set) from the same observer in other images (the negative set) in the dataset, and counting the number of these fixations that fell in areas above the threshold level. The sAUC was then calculated by finding the area under the ROC curve (TPs vs FPs) for each observer-image combination. The sAUCs were averaged across all observers and images to get a final value for each map. The error bars for the sAUCs were obtained from bootstrapping the images and participants for 1000 trials.

**Frequency of fixations/fixation time to object categories.** Each scene had a maximum of 10 bounding boxes corresponding to different objects or regions. These bounding boxes were assigned specific identifiers or categories based on their relevance to scene understanding, as measured by the scene understanding map (SU-relevant and SU-irrelevant), the top predictions of the fixation prediction models (e.g., most salient), and condition-specific identifiers (search target from the object search condition). We counted the frequency of fixations or added the fixation times for these object/region categories across all participants. If an object category contains more than one object for a given image, we averaged fixations/fixation times for these objects.

**Inter-observer correlation of fixation maps.** For each condition, the 25 participants were divided into two random groups of 12 participants each. The correlation of the heat maps for each image was computed between these two groups. This process was repeated for 1000 trials. The average correlation across all images and 1000 trials was used as the inter-observer correlation for each condition. The correlations across Winograd images were computed using 25 participants in each group. In contrast, within-observer correlation computations have half the number of participants in the analysis. We computed the Winograd correlation with half the number of participants to make a valid comparison with the inter-observer correlation. We used 1000 combinations of 12 participants (with non-repeating participants) out of the 25 and calculated the mean fixation heat map correlation for all images for each condition. We performed a similar analysis for across-condition correlations (Fig. 2c), where we take 1000 combinations of 12 participants (with non-repeating participants) for each condition across all images to compute the mean correlations.

**Computing Similarity using Large Language Models (LLMs).** To compute the similarity of experiment participants' descriptions to the gold standard (defined in the methods above), the embeddings of the large language model (Gemini[115]) were utilized. The model API provides access to convert text into learned feature embeddings. A cosine similarity score was used to measure the similarity between these embeddings. To ensure consistency, we implemented the same analyses using the embeddings of another large language model, GPT4[116]. We used cosine similarity to assess the agreement between the LLM similarity metric and the human ratings obtained for all comparisons in the object erasure experiment.

**Computing the lower and upper bound for similarity scores. Upper bound:** Each Winograd image had five gold standard descriptions (defined in the methods above). A pairwise similarity comparison of all five gold standard descriptions was computed using the embedding similarity measure. The average similarity score across all pairwise comparisons across all images constitutes the upper bound for the similarity scores.

**Lower bound:** The gold standard description of each image was compared to the descriptions obtained from the force fixation scene description experiment for another random image in the dataset using LLMs. The lower bound was computed by averaging the similarity scores for 1000 such trials.

**Fixations executed by observers who incorrectly described the scene.** We calculated the fixation frequency distribution for object categories for images with correct and those with incorrect descriptions. The classification of the descriptions as correct or incorrect was based on human judgment similarity ratings relative to the gold standard description, and similarity ratings based on the embeddings of a Large Language Model, Gemini[115].

For the LLM similarity measure, the participants whose description rating fell below one standard deviation (SD) from the mean rating were classified as participants who did not correctly describe the scene, and the rest were grouped as those who correctly described the scene. We also investigated how the analysis varied with different thresholds for categorizing descriptions: SD cutoffs ranging from 0 to 2.5 below the mean similarity rating for each image.

We compared the LLM similarity measures to human classification of correct and incorrect descriptions. The first author and three other research assistants (RAs) from the lab judged the correctness (binary classification: correct or incorrect) of these descriptions (based on the gold standard descriptions). A majority decision across the four raters

was used to assign the final correct/incorrect classification. Ties were resolved through discussion to reach a consensus agreement.

## Statistical analyses

**Data analysis tools.** Python (3.6 or above) and PsychoPy (2020 or above) are used to set up experiments, and other Python libraries are used to handle data (jsonlines, json_lines, pandas). Amazon Mechanical Turk was utilized to conduct online studies. Python libraries like numpy, pandas, and scipy were used for analyzing and testing statistical significance, and matplotlib, cv2, and seaborn were used for plotting and visualizing data and images.

**Bootstrap resampling.** Error bars for all our analyses were obtained using bootstrap resampling of participants, images, descriptions, and ratings. Below, we describe the general procedure. We created 100,000 resamples (with replacement) of images and participants for all the fixation frequency analyses. We used 10,000 observer/image resamples for the fixation heat map correlation analyses because of the computational cost of calculating heat map correlations. In the analysis that quantified inter-observer fixation heat map correlations, we picked 10 random samples of 12 participants from the 25. For each of the 10 samples, we used only 1000 bootstrap resamples ($n = 12$) due to computational constraints. The Cohen's d reported was computed either across the image or the participant-level distributions of the data, depending on the analysis.

**False discovery rate.** All analyses involving moderate to large sets of comparisons had their significance levels corrected for the False Discovery Rate (FDR, $\alpha = 0.05$) using the Benjamini-Hochberg method[117], and the adjusted p-values were reported (q-value). For the cumulative fixation distributions, the comparison across SU-relevant and SU-irrelevant for each successive fixation depended on previous comparisons. Therefore, we added a simple, conservative modification to the FDR procedure, as detailed by the Benjamini-Yekutieli procedure[118]. All significances were determined using a one-tailed significance level.

## Reporting summary

Further information on research design is available in the Nature Portfolio Reporting Summary linked to this article.

## Data availability

The images created as part of the study have been deposited in a Mendeley repository[119] (https://doi.org/10.17632/z6jb259pcd.1). The repository also contains preprocessed data for plotting the fixation distribution across object categories and the cumulative fixation line plots. The code to access and visualize eye movement data is provided in a GitHub repository[60] (https://doi.org/10.5281/zenodo.17374055).

## Code availability

The code to generate SUMs for the images in our dataset is provided in the GitHub repository[60] along with a README file that explains how to access the data and run the code.

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

## Acknowledgements
This study was supported by the Institute for Collaborative Biotechnologies (ICB) cooperative agreement W911NF-19-2-0026, Noyce Foundation (M.P.E.). The views and conclusions contained in this document are those of the authors and should not be interpreted as representing the official policies, either expressed or implied, of the US Government. The US government is authorized to reproduce and distribute reprints for Government purposes, notwithstanding any copyright notation herein.

## Author contributions
The authors, SM and MPE, contributed equally to the design of experiments, data analysis, and writing the paper. SM conducted the experiments and analyzed the data under MPE's supervision.

## Competing interests
The authors declare no competing interests.
