## [Transparent Peer Review file · Nature Communications]

How the Curious Mind Uses Free-Viewing Eye Movements to Maximize Scene Understanding

Corresponding Author: Mr Shravan Murlidaran

Version 0:

Reviewer comments:

Reviewer #1

(Remarks to the Author)

Review for 'The Curious Mind: Eye Movements to Maximize Scene Understanding'

Summary and strengths

This paper addresses a fundamental question about human gaze behaviour: What are people doing when they are 'free viewing', that is, looking at a scene without a specific task or instruction? This question is of particular importance as many (if not most) key publications and databases on human gaze rely on free-viewing data. Previous models assume that people primarily fixate on locally salient or meaningful regions or on objects which share semantic similarity with many other objects in the scene ('concept maps'). The authors propose that instead, people try to understand the scene. This proposal is similar to Loschky's SPECT theory about event comprehension but applies it to the free viewing of static images. Specifically, the authors propose that people fixate those parts of a scene most which matter most for its global understanding.

The paper tests this hypothesis with a clever experimental design, which disentangles local salience and meaning from importance for contextual, global understanding. Stimuli come in pairs of images, in which a given object is once crucial and once irrelevant for scene understanding, because of its relevance for a depicted or inferred action (see Fig 1b of the paper as an example). As predicted under the authors' hypothesis, an object is far more potent in attracting human gaze when the context renders it crucial for the global understanding of the scene. Crucially, this effect is not adequately predicted by any of the previous models but can be captured with 'scene understanding maps'. These maps are produced by iteratively removing objects from a scene and quantifying the changes this removal causes in scene descriptions (either via human ratings or LLM embeddings). The authors further show that this effect cannot be reduced to the objects being perceived to be gazed at or grasped by actors in the images. Finally, the authors run a separate forced fixation experiment with brief presentation times, showing that there is a causal connection between scene understanding and fixating the crucial object. Fixating on the irrelevant object instead leads to erroneous descriptions of the scene.

In our view, there is much to like about this paper. It presents a simple but powerful hypothesis for an unsolved crucial question and presents convincing evidence across a range of predictions and experiments, enabled by a clever design. Given the broad relevance of free viewing data for studies on gaze behavior in psychology, neuroscience, computer vision and clinical fields we think it is a highly important contribution for the broad readership of Nature Communications. Nevertheless, we think a few improvements are possible, which we suggest below.

Suggestions

- How important is scene understanding for typical free-viewing behaviour? The paper convincingly shows that scene understanding is a strong driver for the type of images used in this study. The authors discuss that this may not generalise well to 'empty' scenes not implying any past, present or future actions, which we appreciate. However, it also seems unclear how strong the effect is for other images depicting people and events, which are not designed to imply a crucial role for a single object. Scene understanding maps may capture fixation behaviour which previous models fail to do, for other images of scenes and events as well, but we don't yet know this. This seems a relevant and exciting question, esp. because scene understanding maps are potentially image computable with the help of multimodal transformers. We think this warrants further discussion. To be clear, we do not consider this a problem for the main conclusion of the paper. It's a virtue of the

study that the authors deliberately designed their stimuli to be maximally sensitive to the hypothesized effect and disentangle it from other forms of saliency. However, it's a relevant question for future research how important this effect is relative to other baseline models for everyday scenes, not specifically chosen for this purpose.

- A related and important point is the nature of images and differences between image pairs. We applaud the authors for making the stimuli available but encourage them to show more examples in the paper and describe them in more depth. The current description of 'small' changes between images seems a bit misleading. Changes go beyond an object's position _or_ actor's posture. Typically, there is a whole array of changes to imply a different narrative, including changes which will affect the low-level saliency and local meaning of the object (e.g. the laundry being placed more centrally or the projector being switched on in the example given). This of course is controlled for by the baseline models but seems worth pointing out. Also, some of the scenes seem hard to understand. The analysis focussing on misunderstood scenes used an image-wise deviation from the gold standard description of 1 SD as cut-off. We suspect this may mean very different levels of (mis)understanding for different images. It would be good to provide a feeling for this by showing some example descriptions and discuss the nature of the scenes in this light.

- The discussion points out that models like DeepGaze could in theory pick up on contextual clues but may fail to do so because of the images they are trained on. Another important reason seems that these models are trained to predict fixation density maps, rather than peaks in those maps. The main analysis of the paper uses metrics that these models simply weren't optimised for, and Fig S2 shows that they perform at the same level as scene understanding maps when considering AUCs for the whole map. This doesn't take away from the clear evidence for the importance of scene understanding for free-viewing over and above what these previous models can explain. But the lacking performance of these previous models could be discussed with more nuance. A related question is how the models compare in terms of AUC when not leaving out the fixations attracted by people?

- The abstract states that the 'human brain's default task during free viewing is to understand scenes'. This may be a bit premature, given the study only tested WEIRD adults. There is a plethora of studies showing individual, cultural and developmental differences in free-viewing. These differences may mean that not everybody's gaze is driven by scene understanding. Alternatively, differences in gaze may reflect differences in scene understanding. The latter seems supported by recent findings regarding the connection between individual gaze and individual scene descriptions (Kollenda, Reher & de Haas, 2024). Either way, this point seems worth discussing and phrasing with a bit more caution.

Minor:

- The discussion states the prediction of future events as an important function of scene understanding. A recent paper by Roth et al. (2024 PsychScience) shows that gaze is attracted by perceived potential for change and seems relevant in this context.

- The example image in Fig 1 does not seem to be part of the stim set uploaded to figshare. Was it not used in the experiment?

- Typo on page 14: 'p ! 0.01'

- Fig 4a: The arrangement of the maps is potentially confusing, because the reader is tempted to group them in columns, while there is no one-to-one relationship between models and experimental conditions. Maybe consider a line between the upper (predicted) and lower (measured) rows of maps?

Reviewer #2

(Remarks to the Author)

Summary: The authors investigate the fundamental goal of human eye movements during free viewing. Prior research has suggested that fixations are guided by low-level saliency or local meaning. Here a new "theory" is tested: free viewing seeks to maximize scene understanding by directing gaze toward objects critical to scene interpretation (rather than just salient or locally meaningful regions). The authors introduce Winograd image pairs, where small visual changes change the scene's meaning without changing its saliency or local meaning maps. Eye-tracking experiments across four viewing conditions (free viewing, scene description, object search, and counting objects) reveal that free-viewing fixations closely resemble those during scene description (and are significantly different from object search or counting). Furthermore, observers fixate most frequently on objects that, when removed, alter the scene's overall interpretation—even more than objects identified as the most salient or locally meaningful. A forced fixation experiment confirms that fixating on scene-critical objects enhances comprehension, providing causal evidence that eye movements actively support scene understanding. The study also compares multiple fixation models (saliency, DeepGaze, meaning maps, and the newly developed Scene Understanding Maps (SUMs)), showing that SUMs outperform previous models in predicting human fixations.

Overall Evaluation: This paper presents an interesting contribution, but the fixation data are over-interpreted. It is unsurprising that scene interpretation guides eye movements, and I would hesitate to call this a theory, given that multiple fixations are required to bring high-resolution information to the brain—especially when relevant details are spatially

distributed. Moreover, in small scenes (as shown in the figures), where no fixation is needed to capture the full visual angle, the free-viewing framework collapses, reducing the process to covert attention. That said, the use of Winograd image pairs is a clever approach, effectively disentangling competing explanations and supporting the role of scene interpretation in fixation behavior.

Major concern: A major limitation of the study in relation to the proposed theory is that it removes fixations on people, even though humans are often central to scene interpretation. Faces, gaze direction, and hand actions provide crucial contextual information about events, interactions, and object use, making them integral to understanding a scene's meaning. By excluding these fixations, the study risks underestimating the role of social elements in natural vision and may artificially simplify how scene understanding operates. While this approach isolates object-driven fixations from social biases, it also creates a paradox: if free-viewing supports scene interpretation, ignoring people—the most semantically rich elements in many scenes—limits the generalizability of the findings. A more nuanced analysis could compare fixations in “people-rich” versus “object-rich” scenes or distinguish between fixations on people for social attention versus functional scene understanding. Additionally, understanding when in a sequence of fixations people are looked at would clarify how the visual system prioritizes scene information, offering a stronger theory of scene understanding.

Recommendation: As it stands, this paper would be better suited for a specialist journal such as Journal of Vision or Vision Research rather than a broad-readership publication like Nature Communications. However, if the authors were to revise their theory of scene interpretation to explicitly incorporate social biases in fixations, the paper would become significantly more compelling and more aligned with their own fixation data. This would likely require revisiting their analyses and potentially conducting a control experiment, but I strongly encourage the authors to pursue this approach. Rather than omitting what may be some of the most insightful findings, embedding social fixations into their theoretical framework would enhance both the impact and the generalizability of their conclusions.

Reviewer #4

(Remarks to the Author)

Version 1:

Reviewer comments:

Reviewer #1

(Remarks to the Author)

We congratulate the authors to the revised version of what we consider a ground-breaking contribution.

Even though this point was raised by reviewer 2, we particularly appreciate the analysis including people. In our view, this strengthens and focusses the message of the paper: While people tend to be highly salient in general (and this is a good heuristic for gaze prediction), the authors provide compelling evidence for a novel explanation as to why this is the case. Crucially, this extends to the decisive and much more challenging test case of scenarios in which inanimate objects compete for attention.

It is important to re-iterate that after decades of salience models for static scenes, this is the first compelling evidence we know that human gaze is guided by context-dependent understanding of global meaning. It also provides a convincing answer to the notorious question in the field: Which task are people solving when free-viewing.

The revisions addressed all our points well. Three remaining minor suggestions are:

- The authors may consider adding a concluding paragraph at the end of the discussion which summarises the main message.
- While we understand the reasoning behind using a conceptual, simple example image for Fig 1a, we think it would be good to point out in the figure legend that this example wasn't actually used in the experiments.
- The example images in Figure S5 are small and don't have enough resolution to provide useful information when zooming in. It would be good to replace them with larger versions with higher resolution.

(Remarks on code availability)

Reviewer #2

(Remarks to the Author)

Overall Evaluation: The authors have substantively addressed my main concern about excluding people from the analysis. They rebuilt the Scene Understanding Maps (SUMs) with people deletions included, re-ran all fixation analyses across tasks, added a dedicated results section ("Fixation to people and their role in scene understanding") and new Figure 8. In the revised results, as I expected, people are the most critical element for understanding the scene in ~78% (28/36) of images and attract the most fixations; SU-relevant objects are typically second. These additions bring the theory into alignment with the known prominence of people in natural vision and with the reported fixation data.

Coherence of the revised theory (people vs. objects): The framework now treats fixations as directed to whichever component—person or object—maximizes scene understanding, which coherently explains both frequent fixations on people and systematic fixations on SU-relevant objects. Temporal analyses show that early fixations (1st–2nd) are often drawn to grasp/gaze-cued objects, but SU-relevant objects overtake them by the 6th–7th fixation, while fixations to people remain highest throughout free viewing and scene description. Task differences are sensible (patterns diverge for object search/counting). Overall, this integration is theoretically consistent and empirically supported.

Main comments:

(1) The Introduction (and to some extent the Abstract) are still very much object-centric. This matters because the results show people are the most critical elements in ~78% of images and attract the most fixations—central to the paper's message. The Introduction should be rewritten with the prevalence on people signposted (and not discovered late in the Introduction/deep in the Results section). In addition, clarify the unified prediction in the Introduction/Discussion: fixations maximize scene understanding across components, with people treated as just another (often dominant) component—tying this to the new Figure 8 results.

(2) When people are included, AUROC differences flatten (SUM \approx DeepGaze \approx Meaning Maps; SUM lower, ns). Please surface this prominently and nuance claims of superiority.

(3) The expanded analyses are still post-hoc on the same Winograd-style dataset. A short paragraph in Discussion should acknowledge limits for broader, people-rich everyday scenes and state predictions for a more heterogeneous corpus.

(Remarks on code availability)

Reviewer #4

(Remarks to the Author)

(Remarks on code availability)

Version 2:

Reviewer comments:

Reviewer #2

(Remarks to the Author)

I thank the authors for addressing my final comments. The paper is better balanced as a result. The paper will be a worthwhile contribution to the literature on scene understanding.

(Remarks on code availability)

DEPARTMENT OF PSYCHOLOGICAL & BRAIN SCIENCES

SANTA BARBARA, CA 93106-9660

TEL: (805) 893-2791

FAX: (805) 893-4303

WEB: <http://psvch.ucsb.edu/>

7/04/2025

Dear Reviewers,

We thank the reviewers for the thoughtful comments, which have helped us revise our paper, improve its clarity, add new results, and increase its potential impact.

In general, there was enthusiasm about the paper's potential contributions, but concerns were raised. The most serious concern was the exclusion of fixations on people from the analysis (R2) and the lack of assessment of how eye movements to people and social cues fit within the framework of the theory (eye movements to maximize the understanding of scenes). Below, we address this concern and then cover in detail all other items raised by R1 & R3 (joint reviewing) and R2.

Major Concern: Reviewer 2's request to include fixations on people is arguably the most important concern. First, we'd like to explain why we excluded fixations on people's heads and bodies in our main analysis. Our thinking is that if we included the frequent fixations on people and explained these in terms of the contribution to scene understanding, the reaction would be: "This is obvious! We already know that humans fixate on people in scenes the most, and that could be explained by many theories, not just the observer's goal to understand scenes."

Thus, our analysis focused on predicting the fixations on one of multiple objects in terms of their contribution to scene understanding, which is arguably more challenging than predicting that observers' most frequent fixations are directed at people. Our theory can predict which of the objects is fixated on the most among many objects based on their contribution to understanding the scene.

However, we fully acknowledge the importance of the reviewer's point and motivation: to understand how fixations on people in the scenes fit within the current theoretical framework of eye movement to maximize scene understanding.

To investigate the role of fixations on people in the scenes, we repeated all our analyses but included people in the images. The endeavor included: 1) creating new scene understanding maps (SUMs) by deleting each object and also people from the scenes and collecting scene descriptions to re-generate the scene understanding maps; 2) Repeating all fixation analysis of

our data for the four conditions (free viewing, scene description, objects search, object counting) with the new SUMs including people; 3) Repeating all subsequent analyses: correlation of fixations across conditions (free viewing vs. scene description, vs. object search, and object counting); 4) Assessing the correspondence of different scene component (people, scene understanding (SU) relevant object, perceive grasped/gazed at object, etc.) with the maximum value in the SUMs and the most fixated element in the scenes for the four conditions..

The newly reported results (Figure 8, Supplementary Figure 6) illustrate what R2 suggested in their review: *“how fixations to humans are central to understanding the scene’s meaning.”*

We found that the re-estimated scene understanding maps show that people in the scenes were the most important element (i.e., when deleted, they altered the scene description the most in 77% of the images, even more than the SU-relevant object). Below, we show an example of SUM when including the deletion of people and a comparison of the SUM estimated without including the deletion of people in the scenes (see Reply Figure 1). In addition, we found that people in the scenes attracted the most fixations (see Reply Figure 2). Thus, the common finding of frequent fixations on people is well accommodated within the theoretical framework of eye movements to maximize scene understanding. Based on our definition, people are most often the most critical element in understanding a scene and attract the most fixations. The critical object is the 2nd most important element to scene understanding and is the 2nd most fixated element in scenes.

In addition, R2 wanted to see some insight into the temporal dynamics of fixation selection across various people/social cues and critical objects: *“Additionally, understanding when in a sequence of fixations people are looked at would clarify how the visual system prioritizes scene information, offering a stronger theory of scene understanding.”*

We quantified the frequency of fixations to people, objects perceived to be grasped/gazed at, and objects relevant and irrelevant to scene understanding. We focused on the images for which the perceived grasped/gazed at objects were different from the SU-relevant objects. The findings provide interesting insight into the temporal dynamics of fixation selection. Starting with the 2nd fixation, observers fixate on people in scenes the most. Also, observers’ early fixations (1st and 2nd) are more frequently directed to perceived-to-be-grasped or gazed-at objects than objects relevant to scene understanding. Objects critical to the scene understanding (SU-relevant objects) capture an increasing number of fixations as the exploration progresses and exceed the to-be-grasped or gazed at object after the 6th and 7th fixations, but always below the fixations to people in the scene (see Reply Figure 2). These temporal dynamics are similar for the free viewing and scene description conditions but different for the object search and counting objects conditions. We believe these new results increase the impact of our theoretical framework and methodology (Scene Understanding Maps by deleting objects and people in scenes). We thank R2 for pushing us in this direction.

The revised manuscript's main changes to address R2's concerns include the new results section, “Fixation to people and their role in scene understanding,” and the new Figure 8.

Reply Fig 1 (part of new Figure 8 in the revised manuscript). The top left image and its corresponding scene understanding map (SUM, top middle (with people), and right (without people) are computed by deleting each object at a time and assessing the change in the scene description by human ratings the similarity with the gold standard description. Images below show the fixation heatmaps (across all observers) for each condition, including fixations on people.

Reply Fig 2 (part of new Figure 8 in the revised manuscript). Cumulative fixations per image for different image components: people, scene understanding relevant objects (SU-Relevant Objects), objects perceived to be grasped or gazed at, and scene understanding irrelevant objects (SU-irrelevant objects). The figure also shows a cumulative fixation series on “SU-Relevant including people”, which counts fixations to the most critical component, whether an object or people.

Below, we address all individual reviewer concerns. We use the notation R.X.Y to refer to the reviewer (X) and comment number (Y). Because of the joint review, we refer to R1 and R3 as R1&3.

***R.1&3.1.** How important is scene understanding for typical free-viewing behaviour? The paper convincingly shows that scene understanding is a strong driver for the type of images used in this study. The authors discuss that this may not generalise well to ‘empty’ scenes not implying any past, present or future actions, which we appreciate. However, it also seems unclear how strong the effect is for other images depicting people and events, which are not designed to imply a crucial role for a single object. Scene understanding maps may capture fixation behaviour which previous models fail to do, for other images of scenes and events as well, but we don’t yet know this. This seems a relevant and exciting question, esp. because scene understanding maps are potentially image computable with the help of multimodal transformers. We think this warrants further discussion. To be clear, we do not consider this a problem for the main conclusion of the paper. It’s a virtue of the study that the authors deliberately designed their stimuli to be maximally sensitive to the hypothesized effect and disentangle it from other forms of saliency. However, it’s a relevant question for future research how important this effect is relative to other baseline models for everyday scenes, not specifically chosen for this purpose.*

Response 1&3.1: Thanks for the great summary. Indeed, the reviewers are correct about the possibility of using multi-model large language models (MLLMs) to compute scene understanding maps. When we started this research three years ago, image caption models and MLLMs were not capable of accurately describing our Winograd Images. The rapid development of MLLMs with image segmentation and generative AI provides the opportunity to automate the generation of the Scene Understanding Map (SUM). We are actively working on this. Although the automated SUM is not as accurate as using human photoshopped images, human descriptions, and ratings, we are quickly working on problems and narrowing the gap.

***R1&3.2.** A related and important point is the nature of images and differences between image pairs. We applaud the authors for making the stimuli available but encourage them to show more examples in the paper and describe them in more depth. The current description of ‘small’ changes between images seems a bit misleading. Changes go beyond an object’s position _or_ actor’s posture. Typically, there is a whole array of changes to imply a different narrative, including changes which will affect the low-level salience and local meaning of the object (e.g. the laundry being placed more centrally or the projector being switched on in the example given). This of course is controlled for by the baseline models but seems worth pointing out. Also, some of the scenes seem hard to understand. The analysis focussing on misunderstood scenes used an image-wise deviation from the gold standard description of 1 SD as cut-off. We suspect this may mean very different levels of (mis)understanding for different images. It would*

be good to provide a feeling for this by showing some example descriptions and discuss the nature of the scenes in this light.

Response 1&3.2: We now specify that, in some instances, it meant replacing an object with another one (See Methods, pages 22-23). The reviewers are correct that we have tried to minimize changes in the low-level saliency and local meaning, but in some of the images, there might be some small changes in the objects that we replace. We clarify this in the methods section.

Some of the scenes are more challenging, but we found that a majority of observers (82.78%) accurately described the scenes. We have added this information in the Methods section (page 23).

Regarding the analysis focusing on misunderstood scenes, we have followed the reviewers' request and added image examples in the first part of the supplementary section to give an idea of the descriptions below and above 1 SD (descriptions for all images just below and above 1 SD in Mendeley). In addition, to show the generality of the results (higher fixation numbers for SU-relevant objects for correct vs. incorrect descriptions), the supplementary Figure 5b now also shows fixation frequency using human raters' majority decision to assess whether the description is correct or incorrect. Supplementary Figure 5d also shows how our results generalize to other description similarity deviations cut-offs (0 to 2.5 SD below mean similarity for each image). Together, the results show the generality of the findings and their robustness to various criteria and ways to define correct vs. incorrect descriptions (human and LLM embeddings).

R.1&3.3. The discussion points out that models like DeepGaze could in theory pick up on contextual clues but may fail to do so because of the images they are trained on. Another important reason seems that these models are trained to predict fixation density maps, rather than peaks in those maps. The main analysis of the paper uses metrics that these models simply weren't optimised for, and Fig S2 shows that they perform at the same level as scene understanding maps when considering AUCs for the whole map. This doesn't take away from the clear evidence for the importance of scene understanding for free-viewing over and above what these previous models can explain. But the lacking performance of these previous models could be discussed with more nuance. A related question is how the models compare in terms of AUC when not leaving out the fixations attracted by people?

Response 1&3.3:

The reviewer is correct. DeepGaze's emphasis on fixation density maps might make it less capable of successfully predicting the most frequently fixated object (peak of the map). We have now incorporated the reviewer's observation in the discussion section (page 21). In addition, when we included fixation on people, the AUROC analysis once again shows that the

Deepgaze, Meaning maps, and SUMs are not significantly different from each other, although SUMs were lower (reported in Supplementary Figure 6c).

***R.1&3.4:** The abstract states that the ‘human brain’s default task during free viewing is to understand scenes’. This may be a bit premature, given the study only tested WEIRD adults. There is a plethora of studies showing individual, cultural and developmental differences in free-viewing. These differences may mean that not everybody’s gaze is driven by scene understanding. Alternatively, differences in gaze may reflect differences in scene understanding. The latter seems supported by recent findings regarding the connection between individual gaze and individual scene descriptions (Kollenda, Reher & de Haas, 2024). Either way, this point seems worth discussing and phrasing with a bit more caution.*

Response 1&3.4:

The reviewers bring up an important point. Indeed, individual, cultural, and developmental differences contribute to variability in eye movements during free viewing of scenes. The question is whether this variability implies that the observers are not trying to understand the scene. We argue that the precise components of the scene that are important and contribute to the scene understanding can vary across people. For example, a soccer fan looks at a soccer field scene with a sideline to the side raising a flag and describes it as an offside being called. They might also fixate on the linesman or her/his flag. A person unfamiliar with soccer might describe the scene as a soccer game, and the flag or the lineman will not be critical to understanding the scene, and are less likely to be fixated. Thus, observers try to understand the scene, but their cultural background and individual interests shape their understanding and fixations. Thus, observer or culture-specific SUMs might capture these eye movements. We added a sentence about individual differences (page 22) to the discussion and cited the suggested reference. We also mention cross-cultural differences.

We also followed the reviewer’s point and toned down the abstract. The revised abstract now states: “Thus, we conclude that **one** important default task during free viewing for humans is to understand scenes reflected by frequent eye movements to objects that maximize accurate understanding even when these objects are not to be grasped or gazed at.” If the reviewers would prefer further revisions of this concluding statement, we are happy to modify it further and consider suggested phrasing.

***R1&3.5:** The discussion states the prediction of future events as an important function of scene understanding. A recent paper by Roth et al. (2024 PsychScience) shows that gaze is attracted by perceived potential for change and seems relevant in this context.*

Response 1&3.5: Thanks for pointing to this important relevant reference. We have incorporated the reference on pages 21-22.

R1&3.6: *The example image in Fig 1 does not seem to be part of the stim set uploaded to figshare. Was it not used in the experiment?*

Response 1&3.6: We used this example instead of one from our data set because it is a simple image with few objects. One of the objects is the most salient, the other is the most meaningful (glasses), and the third (the remote control) is the most critical to scene understanding. The images in the experiment had many more objects, making it more challenging to convey the ideas we were trying to communicate. We prefer to use this simpler image that makes it straightforward to explain the difference between the three types of objects.

R1&3.7: *Typo on page 14: 'p ! 0.01'*

Response 1&3.7: Thanks for pointing this out. We have fixed it now.

R1&3.8: *Fig 4a: The arrangement of the maps is potentially confusing, because the reader is tempted to group them in columns, while there is no one-to-one relationship between models and experimental conditions. Maybe consider a line between the upper (predicted) and lower (measured) rows of maps?*

Response 1&3.8: Thanks for the suggestion. We have added a line to separate the two rows.

R2.1: *Overall Evaluation: This paper presents an interesting contribution, but the fixation data are over-interpreted. It is unsurprising that scene interpretation guides eye movements, and I would hesitate to call this a theory, given that multiple fixations are required to bring high-resolution information to the brain—especially when relevant details are spatially distributed. Moreover, in small scenes (as shown in the figures), where no fixation is needed to capture the full visual angle, the free-viewing framework collapses, reducing the process to covert attention. That said, the use of Winograd image pairs is a clever approach, effectively disentangling competing explanations and supporting the role of scene interpretation in fixation behavior.*

Response 2.1:

Thanks for the feedback. We understand that the reviewer is unsurprised that scene interpretation guides eye movements. Perhaps the reviewer belongs to the subfield of scene narratives, and our result might seem less surprising. The paper focuses on the “free viewing” task in which participants are not given any instructions. Factors influencing eye movements during free viewing have been debated for the past two decades, ranging from low-level visual features like saliency to high-level semantic information like objects/ local meaning (Meaning maps). Though our results make intuitive sense that people are trying to understand the scene,

it has been a harder feat to prove this is true because low-level image features and high-level semantic information are often highly correlated. It has been hard to explain what influences fixations concretely. By decoupling low-level saliency from high-level semantic information using the Winograd images, we were able to concretely show that scene understanding does influence eye movements even while freely viewing.

We also believe that our conceptual framework and methodology are novel. In our view, the new framework merits being called a theory (compared to all the theories proposed in cognitive and perceptual psychology, but this is, of course, up for debate and opinion). The idea of measuring the contribution of objects and people in the image to scene understanding by assessing the impact of their removal on scene descriptions is new. The prediction that the most fixations will be directed to the object/person that is most critical to scene understanding is, to our knowledge, also new. Empirical evidence showing the causal influence of these fixations on scene understanding has not been reported previously.

***R2.2. Major concern:** A major limitation of the study in relation to the proposed theory is that it removes fixations on people, even though humans are often central to scene interpretation. Faces, gaze direction, and hand actions provide crucial contextual information about events, interactions, and object use, making them integral to understanding a scene's meaning. By excluding these fixations, the study risks underestimating the role of social elements in natural vision and may artificially simplify how scene understanding operates. While this approach isolates object-driven fixations from social biases, it also creates a paradox: if free-viewing supports scene interpretation, ignoring people—the most semantically rich elements in many scenes—limits the generalizability of the findings. A more nuanced analysis could compare fixations in “people-rich” versus “object-rich” scenes or distinguish between fixations on people for social attention versus functional scene understanding. Additionally, understanding when in a sequence of fixations people are looked at would clarify how the visual system prioritizes scene information, offering a stronger theory of scene understanding.*

Response 2.2:

Please see the Major Concern reply above, in which we address this important point. Our approach used the dataset that we had and directly compared fixation on people and objects. We also showed how incorporating people into our analysis shows that there is no paradox (see R2's comment above). As R2 predicted, people are the most important element in understanding a scene. Our cumulative fixation frequency analysis provides insight into the temporal dynamics of fixation selection to the different components of the scenes, including people, social cues (objects perceived to be grasped or gazed at), and scene understanding relevant objects.

R.2.3. Recommendation: *As it stands, this paper would be better suited for a specialist journal such as Journal of Vision or Vision Research rather than a broad-readership publication like Nature Communications. However, if the authors were to revise their theory of scene interpretation to explicitly incorporate social biases in fixations, the paper would become significantly more compelling and more aligned with their own fixation data. This would likely require revisiting their analyses and potentially conducting a control experiment, but I strongly encourage the authors to pursue this approach. Rather than omitting what may be some of the most insightful findings, embedding social fixations into their theoretical framework would enhance both the impact and the generalizability of their conclusions.*

Response 2.3:

We believe our paper's theoretical framework, conceptual change in thinking, experimental rigor, and innovative methodology for computing scene understanding maps merit publication in a broad readership journal. As a fair comparison, the meaning maps paper (Hayes and Henderson, 2017) that preceded our work and addressed similar questions was published in Nature Human Behavior.

Having said this, we fully recognize the importance of the reviewer's point that extending the work to address the role of fixations to people and social cues would greatly increase the impact. We hope to have provided a useful expansion of the analysis to show how fixations to people work well within the framework of scene understanding maps and present insight into the temporal dynamics of fixation selection to people, social cues, and critical objects. We appreciate R2's comments, which pushed us to do additional analysis and integrate fixation with people in our theoretical framework. The new data provides a broader understanding of how all the components fit together. If there are any additional references that we are missing and that R2 deems appropriate, we are happy to include them.

Sincerely,

Miguel P. Eckstein, PhD
Department of Psychological & Brain Sciences
University of California, Santa Barbara

Shravan Murlidaran, PhD candidate
Department of Psychological & Brain Sciences
University of California, Santa Barbara

DEPARTMENT OF PSYCHOLOGICAL & BRAIN SCIENCES
SANTA BARBARA, CA 93106-9660
TEL: (805) 893-2791
FAX: (805) 893-4303
WEB: <http://psych.ucsb.edu/>

8/28/2025

Dear Reviewers,

We thank the reviewers for their thoughtful new comments, which have helped us further improve our revised paper. Below, we address the remaining individual reviewer concerns and suggestions. We use the notation R.X.Y to refer to the reviewer (X) and comment number (Y). Because of the joint review, we refer to R1 and R3 as R1&3.

R.1&3.1. *The authors may consider adding a concluding paragraph at the end of the discussion which summarises the main message.*

Response 1&3.1: Thank you for the suggestion. We have now added a concluding statement at the end of the discussion. The conclusion reads (see page 23):

“To conclude, our findings suggest that during free viewing of scenes, eye movements are directed to people and objects that are critical to understanding the scene, rather than to regions visually salient or judged to be meaningful. The eye movement fixations serve a perceptual function, as they causally improve the accuracy of scene descriptions. The theory of eye movements to maximize scene understanding and its empirical implementation through scene understanding maps provide a unified account of free viewing of scenes. They predict which objects are fixated on frequently, explain the higher frequency of fixations on people over objects, and frequent fixations on objects that are perceived to be gazed at or to be grasped. Together, our findings suggest that an important default task for the human brain during free viewing is comprehending the visual world.”

R1&3.2. *While we understand the reasoning behind using a conceptual, simple example image for Fig 1a, we think it would be good to point out in the figure legend that this example wasn't actually used in the experiments.*

Response 1&3.2: Thank you for the suggestion. We have added a line stating that the image is not part of our study and is solely used for motivating our study (In the caption of Fig. 1a).

R.1&3.3. The example images in Figure S5 are small and don't have enough resolution to provide useful information when zooming in. It would be good to replace them with larger versions with higher resolution.

Response 1&3.3: Thank you for the suggestion. We have resized the images.

R2.1: The Introduction (and to some extent the Abstract) are still very much object-centric. This matters because the results show people are the most critical elements in ~78% of images and attract the most fixations—central to the paper's message. The Introduction should be rewritten with the prevalence on people signposted (and not discovered late in the Introduction/deep in the Results section). In addition, clarify the unified prediction in the Introduction/Discussion: fixations maximize scene understanding across components, with people treated as just another (often dominant) component—tying this to the new Figure 8 results.

Response 2.1: After re-reading the paper, the reviewer has an excellent point. We have expanded the theory so that it refers to people and objects. We then disclose earlier in the paper that we will focus first on objects and then address the well-documented result of frequent fixations on people. We now first discuss this on page 4 (rather than the last part of the introduction). We have also modified the abstract to reflect that the theory predicts fixations on people or objects that are critical to understanding the scene. We also modified the discussion to reflect that the theory proposes that eye movements are directed to people and objects (page 19, first part of the Discussion Section).

R2.2. When people are included, AUROC differences flatten (SUM \approx DeepGaze \approx Meaning Maps; SUM lower, ns). Please surface this prominently and nuance claims of superiority.

Response 2.2: Thank you for the suggestion. We have added in the discussion section, where we discuss Deep Gaze and Meaning Maps, the null result when people are included in the analysis (pages 20 and 21). Because of this addition, we altered the order of the following two sections: The section titled “The role of fixations on people and social cues in the theoretical framework of scene understanding” and the section on Saliency, Deep Gaze and Meaning Maps (Page 20).

R.2.3. The expanded analyses are still post-hoc on the same Winograd-style dataset. A short paragraph in Discussion should acknowledge limits for broader, people-rich everyday scenes and state predictions for a more heterogeneous corpus.

Response 2.3: The section “Generalization to other image types and influence of individual and cultural differences” discusses the limitations of the Winograd-style image and how results might differ for images that do not contain people or that do not imply past or intended actions. We have now added two sentences discussing people-rich scenes. Although we are not certain

about the results with people-rich scenes, our thinking is that the scene understanding maps might be good at identifying which people are critical to the scene understanding and thus are frequently fixated on, and which people in the scene are less frequently fixated because they are irrelevant to the comprehension of the scene. Of course, this prediction needs to be tested. We have incorporated this sentence on pages 21 and 22.

We thank the reviewers again. The review process for this paper has been a perfect illustration of how the peer-review process can improve the quality and impact of a paper.